# Exploring neurodevelopment via spatiotemporal collation of anatomical networks with NeuroSC

**Noelle L Koonce[1†], Sarah E Emerson[1†], Dhananjay Bhaskar[2], Manik Kuchroo[2], Mark W Moyle[1,3], Pura Arroyo-Morales[1], Nabor Vázquez-Martínez[1], Jamie I Emerson[4], Smita Krishnaswamy[2,5,6,7], William A Mohler[8]\*, Daniel A Colón-Ramos[1,9,10,11]★**

[1]Department of Neuroscience and Department of Cell Biology, Wu Tsai Institute, Yale University, New Haven, United States; [2]Department of Genetics, Yale School of Medicine, New Haven, United States; [3]Department of Biology, Brigham Young University-Idaho, Rexburg, United States; [4]Bilte Co., Ventura, United States; [5]Program for Applied Mathematics, Yale University, New Haven, United States; [6]Department of Computer Science, Yale University, New Haven, United States; [7]Program for Computational Biology and Bioinformatics, Yale University, New Haven, United States; [8]Department of Genetics and Genome Sciences and Center for Cell Analysis and Modeling, University of Connecticut Health Center, Farmington, United States; [9]MBL Fellow, Marine Biological Laboratory, Woods Hole, United States; [10]Wu Tsai Institute, Yale University, New Haven, United States; [11]Instituto de Neurobiología, Recinto de Ciencias Médicas, Universidad de Puerto Rico, San Juan, Puerto Rico

**\*For correspondence:**
wmohler@uchc.edu (WAM);
daniel.colon-ramos@yale.edu
(DAC-R)

[†]These authors contributed equally to this work

## eLife Assessment

NeuroSC is an accessible and interactive tool for streamlined observation of neuronal morphology, membrane contact, and synaptic connectivity across developmental stages in the nematode *C. elegans*. This **important** tool relies on **solid** electron microscopy datasets. This resource will be of high interest to *C. elegans* researchers interested in nervous system wiring and circuit function.

**Abstract** Volume electron microscopy (vEM) datasets such as those generated for connectome studies allow nanoscale quantifications and comparisons of the cell biological features underpinning circuit architectures. Quantifying cell biological relationships in the connectome yields rich, multidimensional datasets that benefit from data science approaches, including dimensionality reduction and integrated graphical representations of neuronal relationships. We developed NeuroSC (*also known as NeuroSCAN,* https://neurosc.net/) an open source online platform that bridges sophisticated graph analytics from data science approaches with the underlying cell biological features in the connectome. We analyze a series of published *C. elegans* brain neuropils and demonstrate how these integrated representations of neuronal relationships facilitate comparisons across connectomes, catalyzing new insights into the structure-function relationships of the circuits and their changes during development. NeuroSC is designed for intuitive examination and comparisons across connectomes, enabling synthesis of knowledge from high-level abstractions of neuronal relationships derived from data science techniques to the detailed identification of the cell biological features underpinning these abstractions.

## Introduction

Neural circuit structure supports function. The underlying image data that yields anatomical connectomes (or wiring diagrams) are typically obtained using volume electron microscopy (vEM) techniques (*Collinson et al., 2023*). Since the first complete connectome was published for *C. elegans* (*White et al., 1986*), these last decades have seen an increase in the generation of vEM datasets, as reviewed in *Kaiser, 2023* and others. The expansion in available anatomical connectomes has resulted from recent advancements in: (1) data generation via automation of EM data acquisition (*Xu et al., 2017*; *Eberle and Zeidler, 2018*; *Zheng et al., 2018*; *Phelps et al., 2021*); and (2) alignment, segmentation, and reconstruction (including recent implementation of AI-driven methods) as reviewed in *Galili et al., 2022*; *Choi et al., 2024* and others. As these developing methodologies continue to improve, they will continue to facilitate the generation of additional connectomes of whole brains and organisms.

The increasing availability of vEM datasets, including the first series of developmental connectomes published for *C. elegans* (*Witvliet et al., 2021*; *Yim et al., 2024*), has highlighted the need for new tools to enable intuitive examination and comparisons across connectomes to promote novel discoveries (*Kasthuri et al., 2015*; *Lichtman et al., 2014*; *Barabási et al., 2023*; *Xu et al., 2021*). It has also underscored the fact that vEM datasets contain a wealth of untapped information that has yet to be fully examined, represented, and integrated for more comprehensive analyses (*Perez et al., 2014*; *Brittin et al., 2021*). For example, vEM datasets enable nanoscale explorations of the underlying cell biological features that govern the properties of neural circuit architectures (*Rivlin et al., 2024*; *Brittin et al., 2021*; *Moyle et al., 2021*; *Witvliet et al., 2021*; *Yim et al., 2024*; *Cuentas-Condori et al., 2019*). Yet most of these cell biological features (cell morphologies, contact profiles, organelle positions, and shapes, etc) are not currently represented in most anatomical connectomes. Quantification of cell biological data results in high-dimensional datasets that require new approaches for their analyses and representations. The advances in vEM data generation and the resulting need for new methodologies in data science and integrated representations of neuronal relationships (e.g. from neuronal positions to neuropil structures) are akin to how advances in genetic sequencing required new methodologies in bioinformatics and new, integrated representations of genomic data (e.g. from gene sequence to gene structure; *Swanson and Lichtman, 2016*). Addressing this gap holds the promise of integrating new knowledge from the fields of cell biology, neurodevelopment, physiology, and systems neuroscience towards explaining how nervous system structure underpins its function.

Most representations of anatomical connectomes have focused on defining neuronal relationships at the level of the chemical synapse (NemaNode; WormWiring; EleganSign; FlyWire; *Witvliet et al., 2021*; *Cook et al., 2019*; *Fenyves et al., 2020*; *Dorkenwald et al., 2023*; *Yim et al., 2024*). While the existence of chemical synapses between neuron pairs is an important feature of neuronal communication, these representations do not capture other neuroanatomical features that also underlie neuron structure and function, including contact sites from adjacent (or nearby) neurons. Recent work in *C. elegans* examined neuronal relationships by quantifying neuron-neuron contact sites to build contact profiles, or contactomes (*Brittin et al., 2021*). Examination of the contactome with data science approaches uncovered structural principles that were not evident from interrogating the synaptic connectome alone (*Moyle et al., 2021*; *Brittin et al., 2021*). These included the existence of higher-order structural motifs and the stratification of neurons (*Moyle et al., 2021*), whose hierarchical assembly during development is guided by centrally located pioneer neurons (*Rapti et al., 2017*). Moreover, integrating neuronal adjacencies (contactome) with synaptic profiles (connectome) allowed for a deeper understanding of the functional segregation of neurons within the stratified neuropil structures (*Brittin et al., 2021*; *Moyle et al., 2021*). Key to achieving this were data science approaches such as Diffusion Condensation (DC) and C-PHATE (*Brugnone et al., 2019*; *Moon et al., 2019*), which resulted in reduced dimensionality of the neuronal relationships, revealing architectural motifs across various scales of granularity, from individual neurons within circuits, to individual circuits within the neuropil. These techniques produced graphs that enabled exploration of these computationally identified groups (*Moyle et al., 2021*). DC/C-PHATE graphs are powerful tools, but they have yet to be integrated to connectomics datasets to enable explorations of the underlying cell biological features. This limits their effectiveness for hypothesis generation and comparative analyses across connectomes.

To address this, we generated NeuroSC (https://neurosc.net/) a tool for exploring neuroarchitectures across vEM datasets via novel representations of the connectome, contactome, and anatomical networks. NeuroSC is an online, open-source platform that facilitates comparisons of neuronal features

and relationships across vEM data to catalyze new insights of the relationships that underpin architectural and functional motifs of the nerve ring neuropil. NeuroSC builds on recent publications in whole-brain EM datasets, integrating the latest set of developmental connectomes (*Witvliet et al., 2021*) and employing data science tools (*Brugnone et al., 2019*; *Moon et al., 2019*) to examine neuronal relationships based on contact profiles. NeuroSC was purposefully developed with a different and complementary goal to existing tools that offer web-based visualization and access to large-scale EM datasets, such as Neuroglancer and Webknossos *Maitin-Shepard et al., 2021*; *Boergens et al., 2017*. The explicit goal of NeuroSC is to provide a platform optimized for examining *neuronal relationships* across connectomic datasets. To achieve this, NeuroSC builds on the segmentations emerging from programs like NeuroGlancer and Webknossos, but with tools tailored to explore relationships such as contact profiles in the context of neuronal morphologies and synaptic positions, and across datasets that represent different animals or different developmental stages. Designed as an open-source and modular platform, NeuroSC is intended to integrate with these existing tools and datasets, supporting a synergistic approach to navigating, analyzing, and deriving meaning from complex connectomic resources.

We demonstrate how these integrated representations of neuronal relationships facilitate comparisons across these connectomes, catalyzing new insights on their structure-function and changes during development. NeuroSC achieves this by addressing three challenges in current neuronal representations: (1) accessibility of specific neuronal cell biological features (i.e. synapses and contacts), (2) integration of features for examining neuronal relationships across anatomical scales, and (3) spatiotemporal comparisons of these features across developmental datasets. These challenges were addressed by (1) creating representations of contact sites and establishing the ability to visualize subsets of synaptic sites; (2) enabling synchronous visualization of neuron morphologies, contacts, and synapses and integrating these cell biological features with algorithmically-generated graphical representations of neuronal relationships; and (3) enabling simultaneous exploration of these relational representations across developmental connectomes. NeuroSC was designed as a suite of tools that facilitates future incorporation of additional datasets and representations with the goal of enabling integrated data exploration beyond the available *C. elegans* connectomes. The NeuroSC-based approaches used here for *C. elegans* could be applicable to other systems as new EM-based datasets and reconstructions become available.

## Results

### Comparing contactome-based relationships using C-PHATE

The adult hermaphrodite *C. elegans* nerve ring is a neuropil of 181 neurons of known identities, morphologies, contact profiles, and synaptic partners (*White et al., 1986*). Even for this relatively small neuropil, representations of a single feature type, such as neuronal contact profiles, constitute over 100,000 data points of multidimensional information: cell identity, region of contact, presence of synapses, etc. Analysis of this multidimensional information requires approaches that can both capture higher order patterns of organization while enabling researchers to access the underlying cell biological features resulting in these relationships. We implemented DC, a clustering algorithm that iteratively groups neurons based on the quantitative similarities of their 'contact' or 'adjacency' profiles (*Brugnone et al., 2019*; *Moyle et al., 2021*). Briefly, DC makes use of pair-wise quantifications of adjacent neuron contacts to, in a graph, move neurons with similar adjacency profiles closer together by applying a diffusion filter in a multidimensional manifold. At each iteration, the diffusion filter smooths the data across the manifold, such that local variability (or noise) in the adjacency profiles is reduced, highlighting broader, higher-order pattern similarities across neurons. As iterations proceed, individual neurons (and eventually groups of neurons) are clustered together based on how close their contact profiles are to one another in the manifold (*Brugnone et al., 2019*). In this way, DC uncovers hierarchical neuronal relationships in the contactome (*Moyle et al., 2021*).

To ensure accurate comparisons of DC across available EM datasets (*Witvliet et al., 2021*; *White et al., 1986*), we empirically determined minimum-distance adjacency thresholds (measured in pixels; *Supplementary file 1*) to construct adjacency profiles. Each neuron's adjacency profile is a quantitative measure that captures the extent of contact with neighboring neurons within its spatial vicinity. By individually setting distance thresholds for each dataset, we ensured that the degree of adjacency—defined by both the presence and extent of contact—could be accurately compared across datasets generated at different times and using diverse methodologies (see also Methods and Materials), schematized in *Figure 1A–C*. We applied an established adjacency algorithm to quantify the

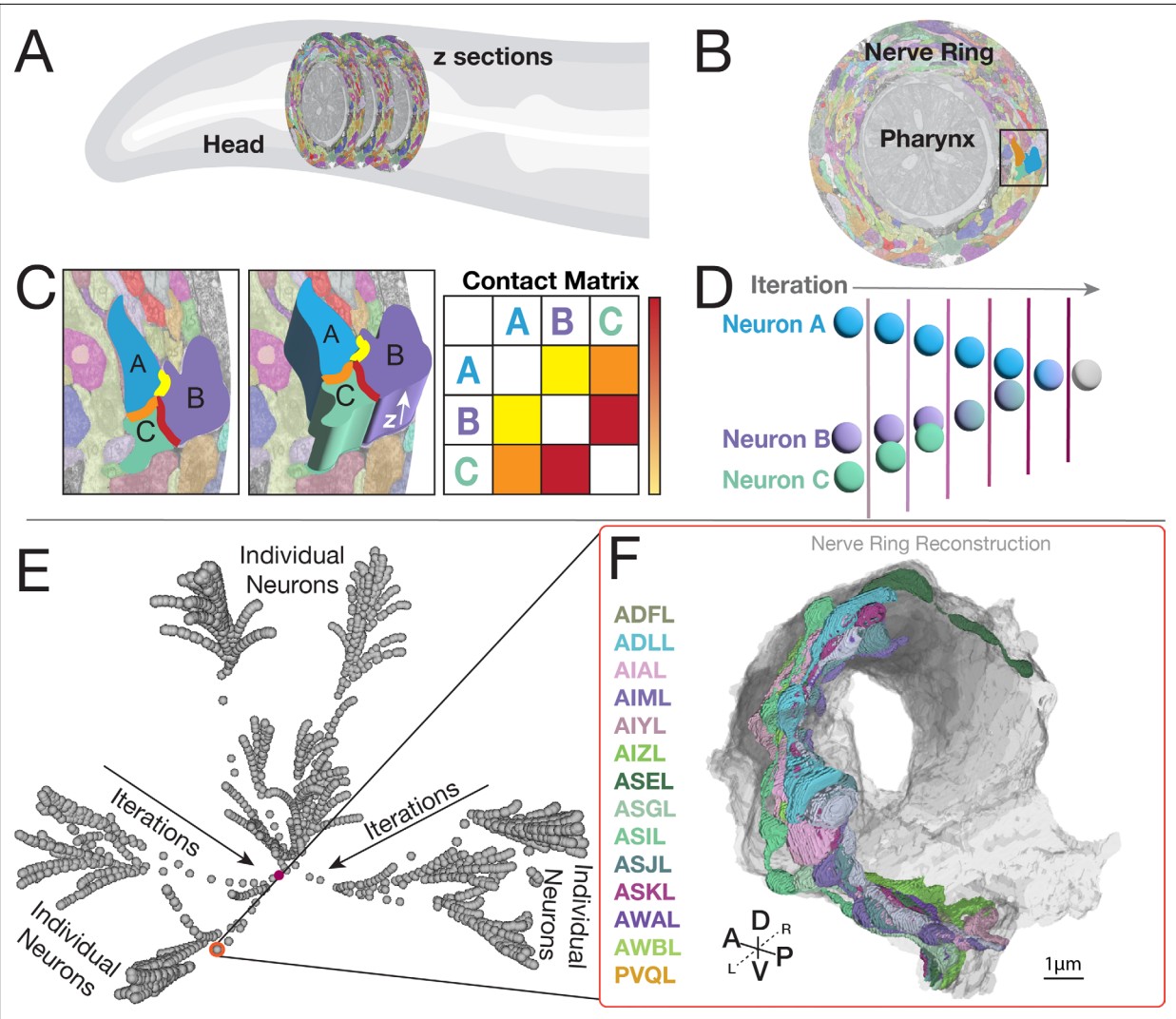

**Figure 1.** DC/C-PHATE representations of contactome-based relationships. DC/C PHATE graphs enable representations of neuronal contact relationships. To build DC/C-PHATE graphs, we (**A**) analyzed serial section EM datasets of the *C. elegans* nerve ring neuropil (located in the head of the animal). (**B**) Single cross-section of the nerve ring (surrounding the pharynx), with segmented neurites pseudo-colored. The dark box corresponds to the zoomed-in image in (**C**). The cross-section is from the JSH dataset digitally segmented (*Brittin et al., 2021*). (**C**) Zoom-in cross section with three arbitrary neurons (called **A**, **B**, **C**) highlighted by overlaying opaque cartoon (2-D, left image) and 3-D shapes (middle image) to represent the segmentation process in the z-axis (arrow) and the neuronal contact sites (highlighted Yellow, orange, and Red). Contacts are quantified for all neuron pairs across the contactome (see Materials and methods), to generate a Contact Matrix (represented here as a table, schematized for the three arbitrary neurons selected and in which specific contact quantities are represented by a color scale and not numerical values). Yellow represents little contact, and red represents a large degree of contact. Here, as an example, you can see that neuron B and C have the largest degree of contact. In an actual contact matrix, this would be a large number of shared pixels. (**D**) Schematic of how the Diffusion Condensation algorithm (visualized with C-PHATE) works. DC/C-PHATE makes use of the contact matrix to group neurons based on similar adjacency profiles (*Brugnone et al., 2019*; *Brugnone et al., 2019*; *Moyle et al., 2021*), schematized here for the three neurons in (**C**). (**E**) Screenshot of the 3-D C-PHATE graph from a Larval stage 1 (L1; 0 hours post hatching;) contactome, with individual neurons represented as spheres at the periphery. Neurons were iteratively clustered towards the center, with the final iteration containing the nerve ring represented as a sphere in the center of the graph (Highlighted in maroon). (**F**) Integration in NeuroSC of the DC/C-PHATE and EM-derived 3-D neuron morphology representations allows users to point to each sphere in the graph and determine cellular or cluster identities for each iteration. Shown here and circled in red, an arbitrarily selected cluster (in E), with the identities of the neurons belonging to that cluster (four letter codes in the column to the left of F) and the corresponding neuronal morphologies (right) of this group of neurons in the EM-reconstructed nerve ring (with individual neurons pseudo-colored according to their names to the left). Compass: Anterior (A), Posterior (P), Dorsal (D), Ventral (V), Left (L), Right (R).

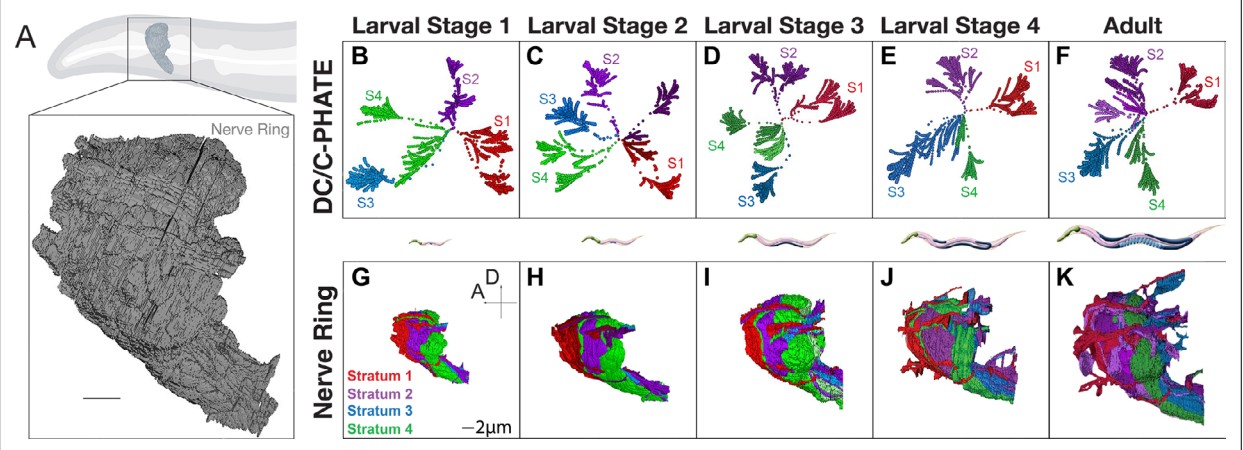

**Figure 2.** Implementation of DC/C-PHATE to developmental contactomes reveals a conserved layered organization maintained during post-embryonic growth. (**A**) Cartoon of the *C. elegans* head and nerve ring (outlined with black box). Below, nerve ring reconstruction from EM data of an L1 animal (5 hours post-hatching), with all neurons in gray. Scale bar 2 μm. (**B–F**) DC/C-PHATE plots generated for available contactomes across *C. elegans* larval development, colored by stratum identity as described (***Moyle et al., 2021***). Individual neurons are located at the edges of the graph and condense centrally. The four superclusters identified and all iterations before are colored accordingly. The identity of the individual neurons belonging to each stratum, and at each larval stage, was largely preserved and is provided in ***Supplementary file 3***; ***Supplementary file 4***; ***Supplementary file 5***; ***Supplementary file 6***. Some datasets contain 5 or 6 super-clusters (colored hues of the stratum that they most closely identify with). These clusters are classified as groups of neurons that are differentially categorized across the developmental connectomes. Note in B the blue cluster extends far to the left due to rotation of the 3D image. (**G–K**) Volumetric reconstruction of the *C. elegans* neuropil from EM serial sections for the indicated larval stages (columns) with the neurons colored based on their strata identity. Scale bar 2 μm; Anterior (**A**) left, Dorsal (**D**) up.

extent of contact between neuron pairs by measuring the number of shared pixels within the defined distance threshold for adjacency (***Brittin et al., 2021***). Pixel counts were summed for each neuron across EM slices within the defined nerve ring region (see Materials and methods), resulting in an adjacency matrix representing pairwise shared pixel counts for each of the seven selected *C. elegans* contactome datasets (L1, 0 hours post hatch (hph); L1, 5hph; L2, 16hph; L3, 27hph; L4, 36hph; Adult 48hph (***Figure 1C***; See also Materials and methods). These adjacency matrices were fed into DC to reveal iterative clusters of neurons with similar adjacency profiles. To visualize and compare the results from DC, we used a graphical representation of the algorithm output called C-PHATE (***Moon et al., 2019***; ***Moyle et al., 2021***), a 3-D visualization tool that builds a hierarchical, visual representation of the DC agglomeration procedure (***Figure 1D–E***). In C-PHATE visualizations, the DC output is mapped in 3-D space with spheres. Initially, all individual neurons in the neuropil dataset are at the periphery of the C-PHATE graph (left-hand side in schematic in ***Figure 1D***, edges of graph in ***Figure 1E***). Neurons are iteratively condensed together based on the similarity of their adjacency profiles (schematized in ***Figure 1D***). In the last iteration of DC, there is a single point at the center of the C-PHATE graph which represents the entire neuropil (***Figure 1E***, red dot). C-PHATE representations enable visualization and comparisons of contactomes across datasets, and explorations of neuronal relationship trajectories, from individual neuron interactions to circuit-circuit bundling (***Figures 1F and 2***).

By Larval stage 1 (L1), 90% of neurons in the neuropil (161 neurons out of the 181 neurons) have grown into the nerve ring and adopted characteristic morphologies and positions. Although the organism grows approximately fivefold from L1 to the adult, contacts in the nerve ring are also largely established by L1 and preserved during postembryonic growth (***Witvliet et al., 2021***). In agreement with this, when we used DC and C-PHATE to examine contactomes from these datasets, we consistently identified four main superclusters: Stratum 1, Stratum 2, Stratum 3, and Stratum 4, using the clusters found at the highest modularity score (the iteration at which the algorithm has the highest clustering confidence) (***Newman, 2006***; ***Figure 2B–F***). DC outputs for each strata across animals can also be inspected using Sankey diagrams (***Supplementary file 3***; ***Supplementary file 4***; ***Supplementary file 5***; ***Supplementary file 6***). These diagrams detail the neuron members at each iteration of DC, allowing the user to derive quantitative comparisons of clustering events. The alignment of the neuronal morphologies of strata members reveals a persistent layered organization to the nerve ring neuropil (***Figure 2G–K***), and the functional identities of the neurons in each stratum suggest that there is spatial segregation of sensory information

and motor outputs (*Moyle et al., 2021*, *Supplementary file 3*; *Supplementary file 4*; *Supplementary file 5*; *Supplementary file 6*). Our findings are consistent with previous studies on the Larval Stage 4 (L4) and adult contactomes (*Moyle et al., 2021*), and support that neurons establish core relationships during embryogenesis and maintain them during postembryonic growth, consistent with previous studies (*Witvliet et al., 2021*). Our findings also demonstrate the utility of DC and C-PHATE analyses in extracting, visualizing, and comparing the structure of the neuropil architecture across contactomes.

Because DC and C-PHATE allow for the examination of relationships at varying levels of granularity, these diagrams also facilitate the interrogation of the architectural motifs that underlie distinct neural strata. A more detailed examination of clusters reveals that while the overall strata are preserved, the underlying neuronal configurations undergo changes during postembryonic growth (*Figures 2B–F and 3*, see:*Supplementary file 3*; *Supplementary file 4*; *Supplementary file 5*; *Supplementary file 6*). Three general features were extracted from these analyses: (1) individual neurons renegotiate their positions in the context of the identified C-PHATE clusters in different developmental contactomes, suggesting developmental changes; (2) the degree of these changes varied across the distinct strata; and (3) the degree of these changes mapped onto strata containing neurons with functions known to require either more or less neuronal plasticity, such as integrative behaviors versus more fixed reflexive behaviors, respectively. For example, Stratum 1, which contains most neurons contributing to shallow reflex circuits, controlling aversive head movements in response to noxious stimuli, displayed the fewest changes among the developmental connectomes (*Figure 3B–F*; *Supplementary file 3*). On the other hand, *C. elegans* exhibit tractable behaviors which can adapt due to changing environmental conditions *Flavell et al., 2020*. Strata 3 and 4 contain most neurons belonging to circuits associated with such learned behaviors, including chemo, mechano, and thermo sensation. This is reflected by strata 3 and 4 being the regions of the most change in neuronal relationships across postembryonic development (*Figure 3G–L*; *Supplementary file 5*; *Supplementary file 6*).

To examine the changes in DC/C-PHATE during postembryonic development, we made the C-PHATE plots fully interactive. This enables users to hover over and identify members of each intermediate cluster, to highlight specific cell trajectories via pseudo-coloring, and compare specific neuronal relationship dynamics across development within a multi-view window of distinct C-PHATE plots (*Figure 1E-F*, *Figure 8—figure supplement 2*; *Figure 4—video 1*). Because C-PHATE graphs ultimately represent cells of known identities, we reasoned that interactive mapping of the C-PHATE cluster objects to their component cellular identities and anatomies could yield greater insights on

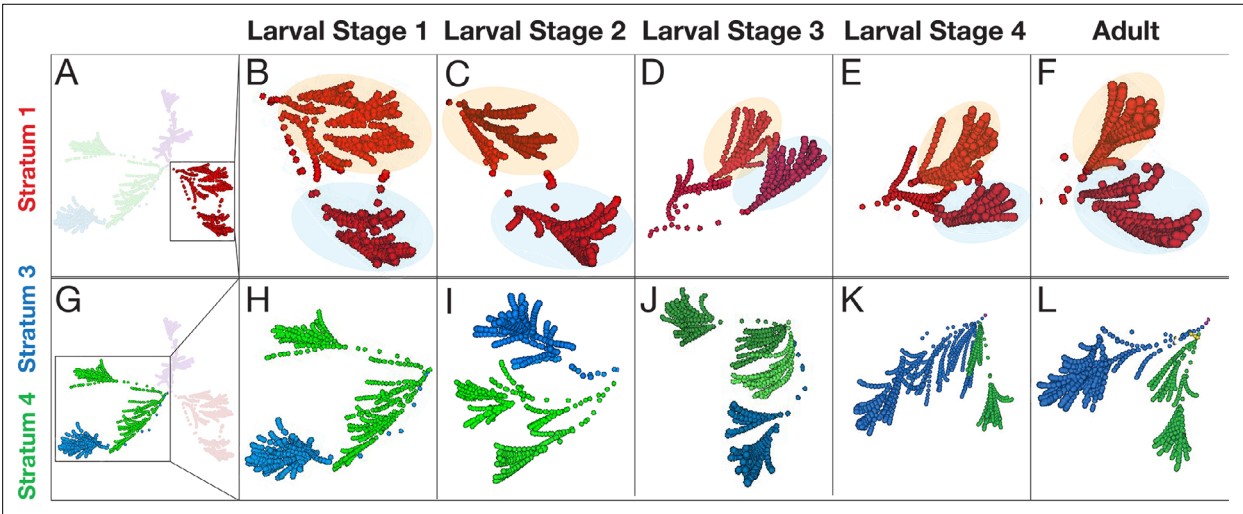

**Figure 3.** Examination of the architectural motifs underlying the distinct strata across development. Visualization of (**A–F**) Stratum 1 (Red) and (**G–L**) Strata 3 and 4 (Blue and Green) reveal motifs that are preserved (Stratum 1) and change (Strata 3 and 4) across developmental contactomes (L1 to Adult, left to right, as indicated by labels on top). (**B–F**) Cropped view of Stratum 1 at each developmental stage showing a similar shape of two 'horn-like' clusters in the C-PHATE graphs (as seen by orange and blue shaded areas). These two clusters have similar neuronal memberships, which are largely invariant across developmental contactomes (*Supplementary file 3*). (**H–L**) Cropped view of Strata 3 and 4 at each developmental stage highlighting differences in the organization and number of neurons contained in each of the Blue and Green strata, which is particularly distinct when comparing (**H**) L1 and (**K**) L4 (*Supplementary file 5*; *Supplementary file 6*).

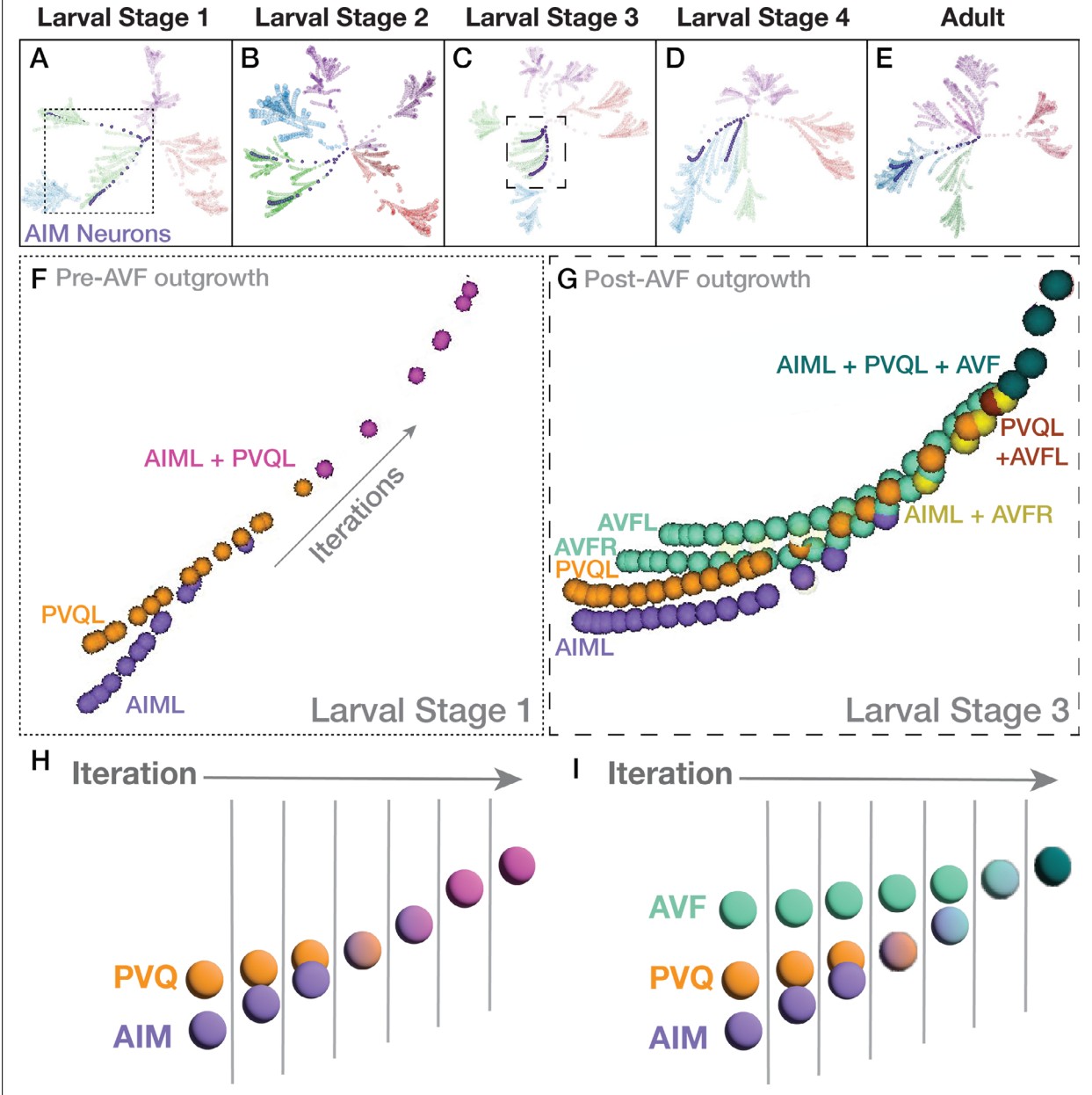

**Figure 4.** Case study: AIML and PVQL neurons change clustering patterns across the developmental contactomes. (A–E) C-PHATE plots across development, with the trajectories of AIM neurons (in purple) and the rest of the spheres colored by stratum identity (see **Figure 2**). (F–G) Zoom in of the AIML and PVQL trajectories corresponding to larval Stage 1 (pre-AVF ingrowth) (A, dotted box) and in (G), Larval Stage 3 with AVFL/R present (C, dashed box). Note how the relationship between AIM and PVQ neurons in the C-PHATE graph varies for each of the examined contactomes across development. (**Figure 4—figure supplement 1**, **Supplementary file 7**). (H,I) simplified schematics of F and G based on neuron class.

The online version of this article includes the following video and figure supplement(s) for figure 4:

**Figure supplement 1.** DC/C-PHATE clustering of AIM, PVQ, and AVF across postembryonic development.

**Figure 4—video 1.** Visualization of hierarchical relationships using C-PHATE plots in NeuroSC.

https://elifesciences.org/articles/103977/figures#fig4video1

neurodevelopmental changes by linking the algorithmic abstractions of the relationships with the cell biological features and their changes across development (**Figure 4**).

To evaluate our hypothesis and assess the utility of C-PHATE for discovery, we examined specific regions where the distribution or 'shape' of super clusters changed across developmental contactomes. This approach accounts for differences in contact profiles, which directly impacts the overall

clustering structure. Based on these criteria, we focused on a region displaying changes in Strata 3 and 4, and using the interactive C-PHATE graphs (*Figure 4A–E*), we determined the identities of neurons that changed clustering patterns across the developmental contactomes. Specifically, we focused on two interneurons, named AIML and PVQL, which we observed undergo a change in their cluster assignment from Stratum 4 (at L1) to Stratum 3 (at Larval stage 4, L4; *Figure 4A and D*). We pseudo-colored the trajectories of the AIML and PVQL neurons in C-PHATE to explore the changes in their C-PHATE trajectories throughout the developmental stages (*Figure 4F-I*, *Figure 4—figure supplement 1*, *Supplementary file 7*). Comparison of the identities of the neurons that co-cluster with AIML and PVQL suggests that the contact relationships varied across developmental stages (*Figure 4F-G*, *Figure 4—figure supplement 1*), co-cluster members can also be evaluated via Sankey diagrams, showing a switch in membership of AIM and PVQ from clusters of stratum 4 at L1-L2, pre-AVF ingrowth, to include a transitionary stratum at L3, finally to stratum 3 at L4-adult after AVF ingrowth (see *Supplementary file 7*).

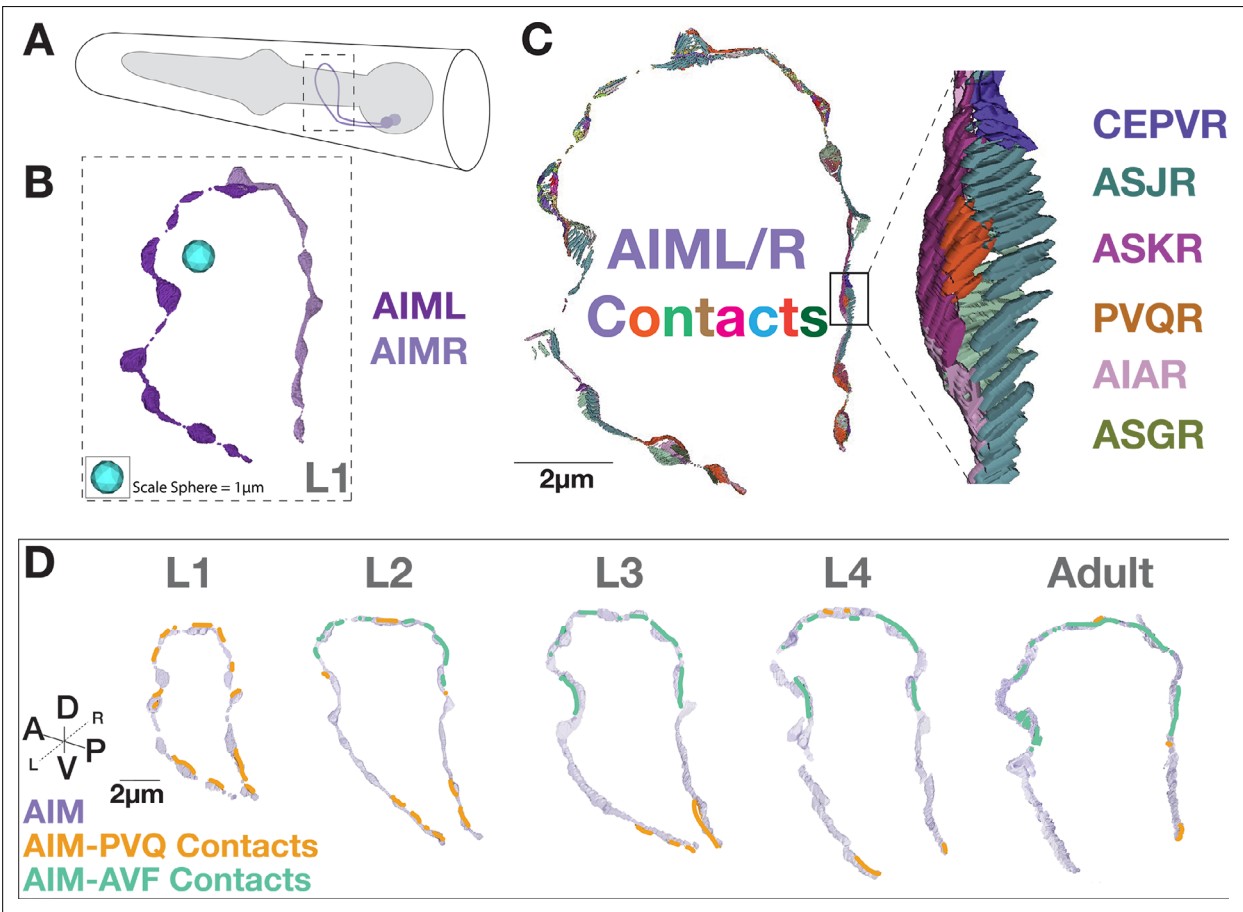

**Figure 5.** Case Study: Visualization of contact profiles in individual neurons. (**A**) Cartoon schematic of the head of the animal with the AIM neurons (purple) and pharynx (gray), and (dotted box) a 3-D reconstruction of the AIM neuron morphology from the L1 (0 hours post-hatching) dataset. (**B**) AIML and AIMR neurites rendered in 3D from L1. Note that we did not implement any surface smoothing methods to objects, so there might be gaps in the renderings. This was done intentionally, with the goal of producing the most accurate representation of the available data segmentation and avoiding any rendering interpretations. (**C**) 3-D representation of all contacts onto the AIM neuron morphology in an L1 animal, colored based on contacting partner identity, as labeled (right) in the detailed inset (black box) region. (**D**) AIM-PVQ contacts (in orange) and AIM-AVF contacts (in green), projected onto the AIM neurons (light purple) across developmental stages and augmented for clarity in the figure (see non-augmented contacts in *Figure 8— figure supplement 1*). Scale bar 2 μm.

The online version of this article includes the following figure supplement(s) for figure 5:

**Figure supplement 1.** Projecting contact profiles onto the segmented neuronal shapes.

**Figure supplement 2.** AIM contact sites.

## Visualizing contact profiles in individual cells

DC/C-PHATE changes should result from changes in contact profiles. To link the observed changes in the C-PHATE graphs with the cell-biological changes in contact profiles, we generated a tool that would simultaneously enable: (1) 3D visualization of the cell-cell contact sites onto individual neuronal morphologies; (2) examination and comparisons of these contact profiles throughout development for the available contactomes; and (3) integration with DC/C-PHATE to link C-PHATE cluster objects to the 3-D morphologies of the algorithmically clustered cells. With these capabilities integrated, we could simultaneously view the contactome from two complementary perspectives – at an abstract systems level via DC/C-PHATE and at a cell biological level via 3D contact modeling – to perceive the architectural themes that underlie similar network patterns.

To create this tool, we generated 3D models of the area of physical contact between adjacent neuron pairs (*Supplementary files 1 and 2*; Materials and methods; *Figure 5*); *Figure 5—figure supplement 1*. Visualizing contacts from all adjacent neurons builds a multi-colored skeleton of the neuron morphology mapped onto the boundaries of this neuron (*Figure 5A and C*). Because the identities of the neurons are known and linked to the 3D contact models, we built text pop-ups that define the contact partners for each site (*Figure 5C*). Furthermore, since neuron names are consistent across the EM datasets, we can link and compare contact sites throughout development (*Figure 5D*). Additionally, we can analyze the representations of contact sites in the context of DC/C-PHATE clustering profiles (*Figure 4F–I*), 3D models of neuronal morphologies (*Figure 1F*), and 3D models of synaptic sites for any neuron(s) across development (Figure 7).

We used the integrated tools of DC/C-PHATE and 3D representations of the contact profiles to examine the potential cell biological changes leading to the DC/C-PHATE clustering changes observed for the AIML neuron during development. With these tools, we observed changes in the identities of the contacts made in the dorsal region of the AIML neurite (*Figure 5D*; *Figure 5—figure supplement 2*). Specifically, in the L2 stage (as compared to L1), we observed a decrease in the contacts from PVQL and an increase in contacts from the AVF neurons. This change persists to the adult stage (*Figure 5D*; *Figure 5—figure supplement 2*).

To then determine the possible source of these developmental changes in contacts, we visualized 3D models of the segmented morphologies for these neurons across L1 to adulthood (*Figure 6*). We find that AIM and PVQ neurons maintain similar morphologies throughout development (*Figure 6C*), while AVF neurons undergo substantial neurite outgrowth onto new regions of contact between AIM and PVQ (*Figure 6B–D*). These observations are consistent with lineaging studies which demonstrated that AVF neurons are generated from neuronal precursors (P0 and P1) at the end of the L1 stage (*Sulston et al., 1983*; *Sun and Hobert, 2023*; *Poole et al., 2024*; *Hall and Altun, 2008*; *Sulston and Horvitz, 1977*). By comparing the EM datasets across development, we observe that the AVF neurons grow into the nerve ring during the L2 stage and continue to grow until the Adult stage (*Figure 6B–D*). We also consistently observe throughout the individual datasets that the AVF neurons grow in between the AIM and PVQ neurons (*Figure 6D*), altering their contact profiles, which likely contributes to the observed changes in the C-PHATE graphs. While the DC/C-PHATE representations systematically cluster neurons based on relative similarities across contact profiles, and not solely by scoring changes in specific contacts within any given pair, our findings demonstrate the use of DC/C-PHATE as a discovery tool to identify cell-biological contact changes during development. (*Figure 4F and G*; *Figure 5D*; *Figure 6—video 1*). Consistent with this, we also observe that both AVFL and AVFR grow into the nerve ring alongside AIML, later continuing to grow around to reach AIMR, and that these relationships were reflected in the C-PHATE graphs in terms of the clustering profiles throughout development (*Figure 4G*; *Figure 4—figure supplement 1*).

We then examined if the developmental changes in contact profiles result in changes in circuitry. We examined this by layering on synaptic information. Despite dwindling AIM-PVQ contacts, AIM and PVQ neurons maintained their synaptic relationship throughout development, with synaptic sites observed primarily at the base of AIM neurons, a region of persistent contact with PVQ (*Figure 7A-B*). We observed that increases in contacts between AIM and AVF neurons resulted in additional en passant synapses at the new points of contact, beginning at the L2 stage and continuing to adulthood (*Figure 7A–B*). We also observed that AVF forms synapses with the adjacent PVQ neurons (*Figure 7*; *Figure 7—figure supplement 1*).

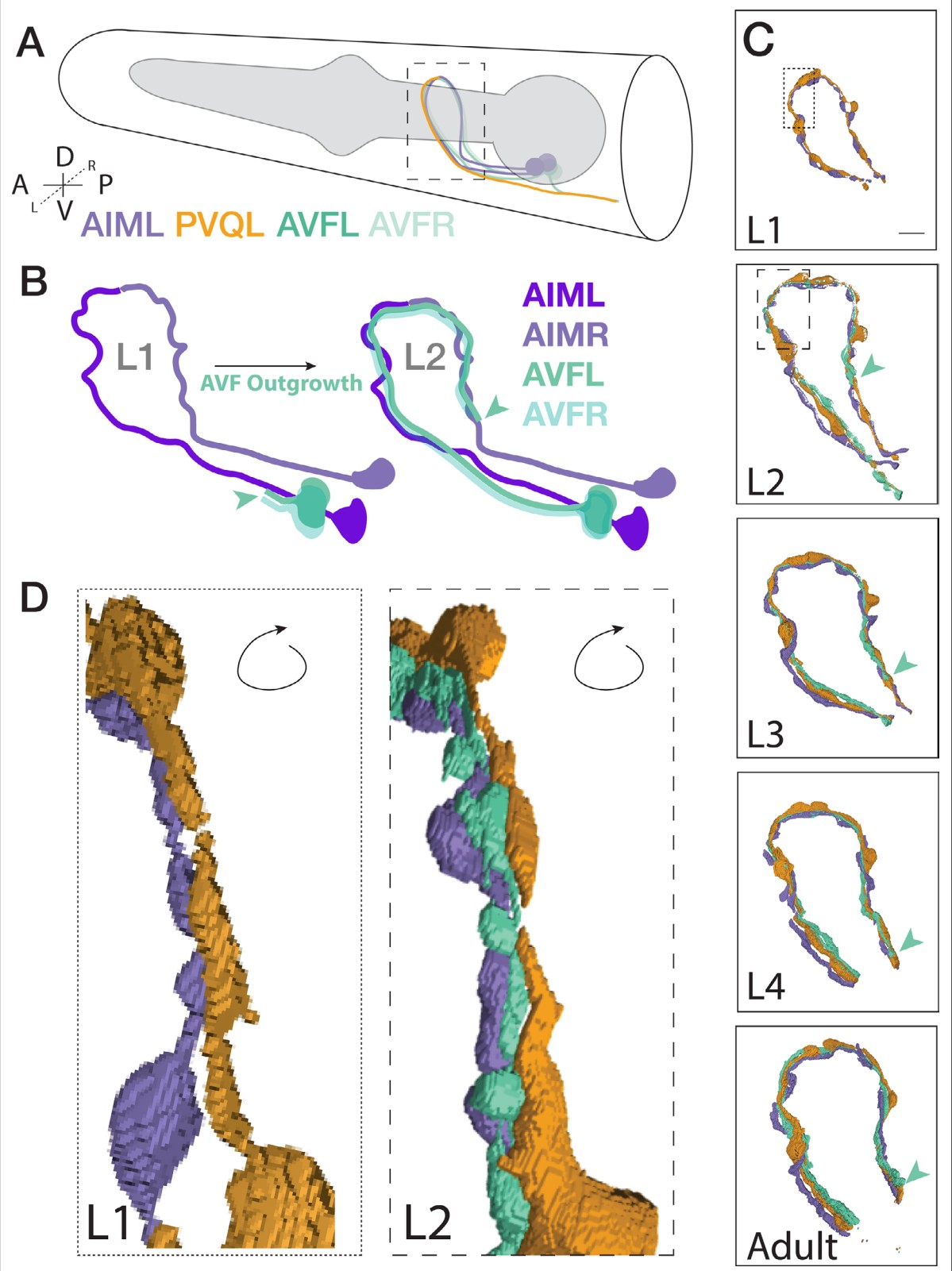

**Figure 6.** Case study: Segmented morphologies of AIM, PVQ, and AVF across larval development. (**A**) Cartoon schematic of the *C. elegans* head, pharynx (gray), and examined neurons with dashed black box representing the nerve ring region. (**B**) Schematic representation of the outgrowth path of the AVF neurons as observed by EM (*Witvliet et al., 2021*). The distal end of the AVFL neurite is highlighted with a green arrowhead in the schematic. (**C**) Neuronal morphologies of AIM (purple), PVQ (orange), AVFL (green) across postembryonic development, as indicated, with green arrowhead

*Figure 6 continued on next page*

*Figure 6 continued*

pointing to AVF outgrowth tip. Scale bar = 2 μm. Regions for insets (L1, dotted box; L2, dashed box) correspond to (**D**). Note that the AVF neuron class is comprised of a left and right counterpart. In C and D, we only show AVFL for simplicity. During development, AVFL and AVFR, as shown in A and B, both grow ipsilaterally along AIML in parallel and extend around the nerve ring together. This is unusual among classes of nerve ring neurons. (**D**) Morphologies of these neurons (rotated to the posterior view) display the AVFL neuron positions between the AIM and PVQ neurons at the L1 and L2 stage. Indicated outgrowth between neurons continues to the adult stage (*Figure 6—video 1*). Note how AVF outgrowth alters contact between PVQ and AIM (*Figure 5D*).

The online version of this article includes the following video(s) for figure 6:

**Figure 6—video 1.** Analysis of AIM, PVQ, and AVF neuronal morphologies in developmental datasets.

https://elifesciences.org/articles/103977/figures#fig6video1

**Figure 6—video 2.** Navigating NeuroSC features that enable integration of Neurons, Contacts, and Synapses across developmental datasets.

https://elifesciences.org/articles/103977/figures#fig6video2

---

In summary, by integrating, representing, and comparing datasets using the new C-PHATE tools and contact profiles in NeuroSC, we identified developmental changes in the relationships of AIM, AVF, and PVQ. This case study highlights the utility of combining cell biological representations (such as morphologies, contacts, and synapses) with coarse-grained systems-level representations (like DC/C-PHATE) of vEM datasets to uncover developmental changes that could be further explored

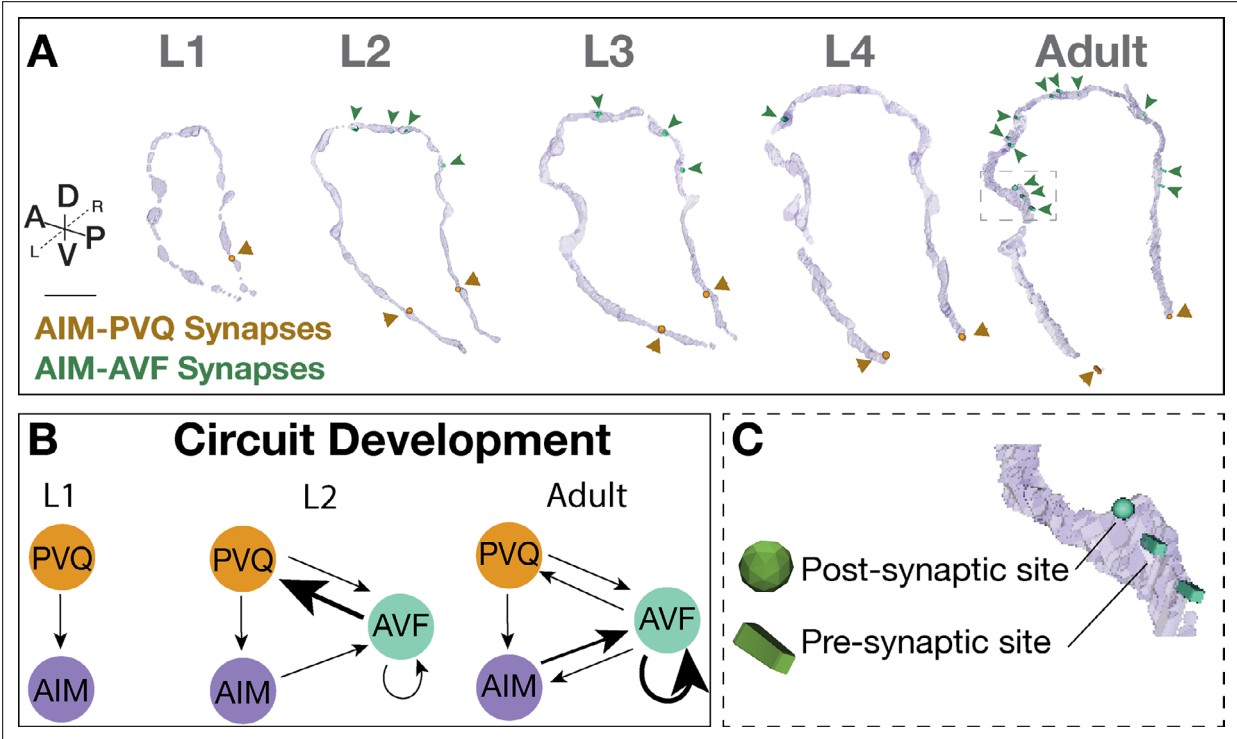

**Figure 7.** Case study: AIM-PVQ and AIM-AVF synaptic positions across development. (**A**) AIM-PVQ synaptic sites (dark orange arrowheads) and AIM-AVF synaptic sites (dark green arrowheads) in the segmented AIM neurons and reconstructed across postembryonic development from original connectomics data. Scale bar = 2 μm. (**B**) Schematic of the AIM, PVQ, and AVF circuitry across development based on synaptic connectivity and focusing on the stage before AVF outgrowth (L1), during AVF outgrowth (L2), and Adult; arrow direction indicates pre to post synaptic connection, and arrow thickness indicates relative number of synaptic sites (finest, <5 synapses; medium, 5–10 synapses; thickest, 11–30 synapses). (**C**) Zoom in of synaptic sites (green) in the Adult connectome and embedded into the AIM neuron morphology (light purple). In NeuroSC, presynaptic sites are displayed as blocks and postsynaptic sites as spheres, and a scaling factor is applied to the 3-D models (Materials and methods).

The online version of this article includes the following video and figure supplement(s) for figure 7:

**Figure supplement 1.** AVF synaptic sites.

**Figure 7—video 1.** Exploring contacts and synapses using NeuroSCAN.

https://elifesciences.org/articles/103977/figures#fig7video1

experimentally. Therefore, NeuroSC serves as a powerful platform for generating hypotheses for empirical testing, which can lead to insights into the dynamics of circuit development.

## NeuroSC: facilitating multi-layered interrogation of neuronal relationships in the *C. elegans* nerve ring throughout larval development

NeuroSC is built as a web-based client-server system designed to enable the sharing of anatomical connectomics data with an emphasis on facilitating the analyses of neuropil relationships across hierarchies and scales. To achieve this, we integrated tools of neuroanatomical investigation from the available *C. elegans* nerve ring connectomes and contactomes with a collection of 3-D modeled elements (morphologies, contacts and synapses and C-PHATE) representing different aspects of neuronal architecture and relationships (*Figure 8*). NeuroSC differs from other available web-based tools in this area with the integration of C-PHATE graphs that enable exploration of hierarchical organizations of stratified fascicles, the availability of new tools to examine the contactome, and the integration of these data with existing connectome and morphological datasets across developmental stages.

NeuroSC has eight key user-driven features: (1) C-PHATE, with the ability to highlight clusters containing neurons of interest (*Figure 8—figure supplement 2*; *Figure 4—video 1*), (2) interactive reconstructions of neuronal morphologies (*Figure 8—figure supplement 6*; *Figure 6—video 2*), with click-based display of cell statistics, including total volume and surface area within the defined neuropil region (see Materials and methods), (3) reconstructions of neuronal morphologies of C-PHATE cluster members with a right-click on C-PHATE clusters (*Figure 4—video 1*), (4) 3-D renderings of neuronal contacts to visualize the spatial distribution of contact profiles, with click-based displays of quantitative statistics, including total contact area, rank among all contacts of the primary neuron, and surface area percentages relative to both the neuron and the nerve ring (*Figure 8—figure supplement 1*; *Figure 4—video 1*), (5) 3-D representations of synaptic sites with the option to visualize subsets of those sites, and a click-based display showing the number of synapses of the selected identity within the primary neuron, including any polyadic synapse combinations involving the primary neurons (*Figure 8—figure supplement 3*; *Figure 7—video 1*) (6) the ability to perform side-by-side comparisons across development by cloning a rendered scene across time into a new window at the click of a button using the in viewer developmental stage slider (*Figure 8—figure supplement 7*; *Figure 6—video 2*). The cloning feature uses the original search query from the first viewer. 'Contacts' may appear or disappear when the user examines different developmental stages if the original query contacts are lost (or gained) at the specific developmental stage in the underlying data. (7) The option to pseudo color each object to highlight points of interest (*Figure 8—figure supplement 7*; *Figure 6—video 2*) and (8) each item is an individual object with the ability to be further customized by the user (*Figure 8—figure supplement 7*, *Figure 8—figure supplement 8*).

## NeuroSC: practical considerations

We offer seven practical considerations for users. First, NeuroSC is available on mobile platforms as a quick and convenient way to look up neuron morphologies and relationships. Second, since contact sites offer the ability to explore the surrounding neurons and the position(s) of contact between adjacent neurons, NeuroSC is designed to enable studies of adjacent neurons (e.g. phenotypes that result in site-specific ectopic synapses; neuron morphology changes that may affect specific surrounding neurons; developmental events requiring communication between neurons, etc.). Third, C-PHATE can be used to identify neurons with similar contact profiles. Because contact profiles are associated with circuit identities (*Moyle et al., 2021*), exploration of neuronal relationships via C-PHATE can be used to identify new relationships between specific neurons and circuits. Fourth, visualization of subsets of synaptic and contact sites allows direct comparisons to light microscopy approaches such as cell-specific labeling of synapses or GFP-Reconstitution across synaptic partners (*Feinberg et al., 2008*). Fifth, because the color and transparency of each 3-D model can be customized, users can further integrate NeuroSC outputs of additional atlases for gene expression, neurotransmitter and receptor expression, functional connectivity, etc. (*Packer et al., 2019*; *Taylor et al., 2021*; *Wang et al., 2023*; *Fenyves et al., 2020*; *Randi et al., 2023*; *Maitin-Shepard et al., 2021*) and directly use the NeuroSC outputs to create figures and comparisons (as done for this paper). Sixth, although synaptic sites with BWM (body wall muscles) are included in NeuroSC, the current data model limits the ability to search

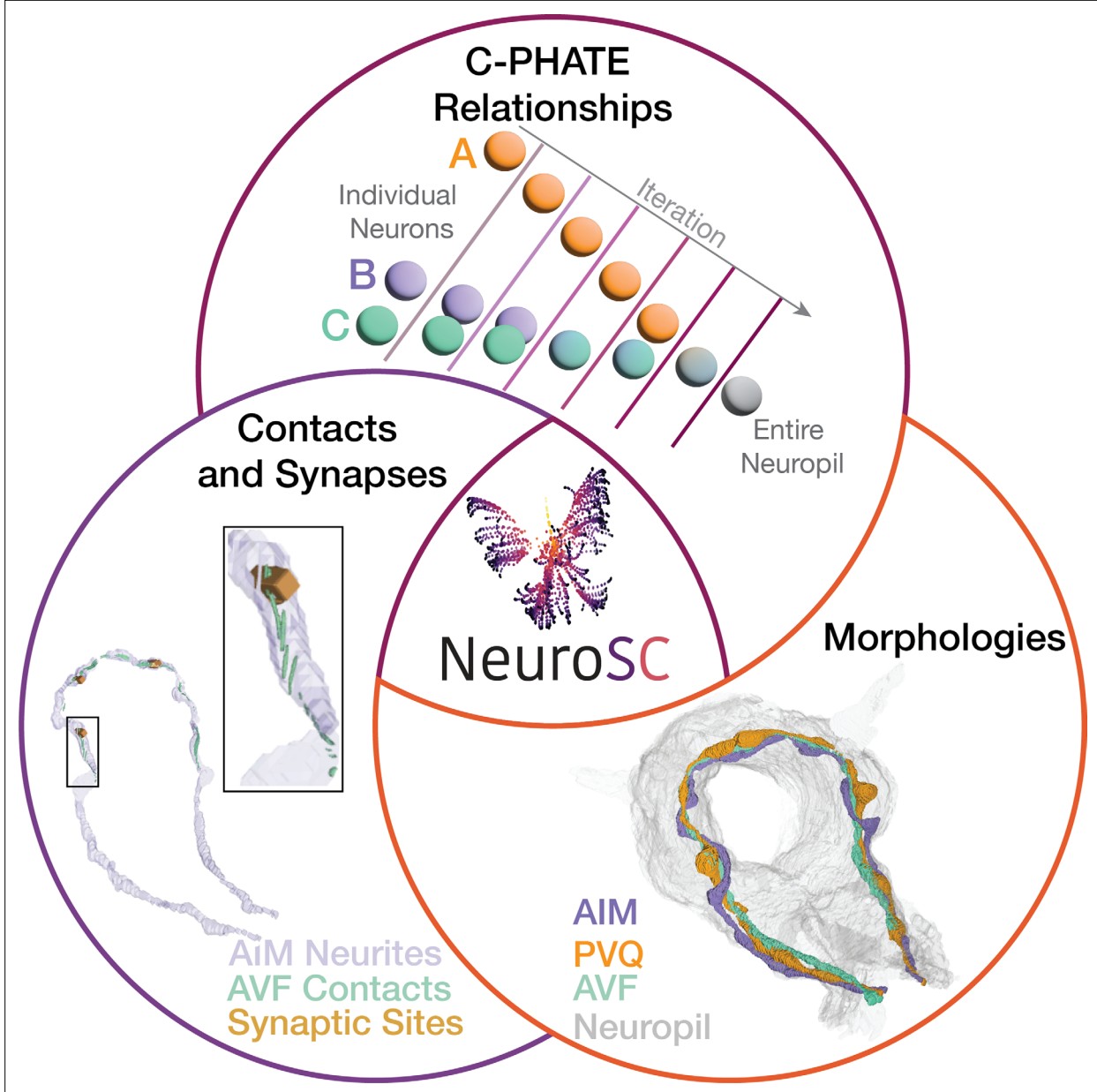

**Figure 8.** NeuroSC is a tool that enables integrated comparisons of neuronal relationships across development. With NeuroSC, users have integrated access to: C-PHATE plots, 3-D morphological renderings, neuronal contact sites, and synaptic representations. Through stage-specific C-PHATE renderings, users can explore neuronal relationships from high dimensional contactome data. (Top) On C-PHATE plots, schematized here, each sphere represents an individual neuron, or a group of neurons clustered together during algorithm iterations. (Right) 3D renderings of select neurons can be visualized in the context of the entire nerve ring or other circuits (gray). (Left) AIM contact sites at L1 and the same region showing synapses. The inset shows a zoomed-in view of contacts and synapses - presynaptic sites (blocks) and postsynaptic sites (spheres). Data depicted here are from the L1 stage (0 hours post hatching).

The online version of this article includes the following figure supplement(s) for figure 8:

**Figure supplement 1.** Visualization of contact sites in NeuroSC.

**Figure supplement 2.** C-PHATE tutorial in NeuroSC.

**Figure supplement 3.** Visualization of synaptic sites with NeuroSC.

**Figure supplement 4.** Opening page view and menu.

**Figure supplement 5.** The NeuroSC interface enables interrogation of neuronal relationships across development.

**Figure supplement 6.** Select and Add objects to viewers.

*Figure 8 continued on next page*

for these non-neuronal cells. Users can search for neurons with synapses to BWM to find this datatype. Seventh, to enable direct comparisons between our data representations and the primary EM data, the original annotations have been preserved and can be accessed by users via the sister app, Cyto-SHOW (CytoSHOW.org). As the data continues to be curated, the modular design of NeuroSC and its companionship with CytoSHOW enables integration of future annotations.

## Discussion

NeuroSC is an integrative tool for analyzing detailed, web-based representations of neuronal connectomes and contactomes throughout post-embryonic development in *C. elegans*. Connectomes and contactomes are derived from volume electron microscopy (vEM) micrographs of neuropil regions (*Witvliet et al., 2021*; *White et al., 1986*; *Yim et al., 2024*). These EM micrographs are information-rich and have the potential to reveal architectural motifs across scales, from the nanoarchitecture of the neuron to the neuroanatomy of each circuit in the brain. Cell biological features, such as contact profiles and synaptic positions, can be rigorously quantified and systematically represented as graphs capturing multidimensional relationships. These representations require methodologies from data science that enable dimensionality reduction and comparisons of the architecture across scales. Yet to derive new intuitions about the spatiotemporal events leading to the architecture that shapes its function, it is necessary to integrate and compare these various representations, bridging knowledge from the cell biological events to the systems-level network relationships. NeuroSC is designed to achieve this integration, enabling synthesis of knowledge ranging from the abstractions of neuronal relationships in C-PHATE to the cell biological features underpinning these abstractions. We provide a case study to illustrate how integration of analyses performed in NeuroSC can result in new insights. First, we demonstrated the discovery process with C-PHATE representations to identify neurons that undergo changes in their contactome during development. Second, we developed 3-D representations of contact sites to analyze the local neuronal regions that were identified via DC/C-PHATE analysis. Third, we visualized and compared these representations across development to identify cell biological changes in neuronal morphologies and synaptic positions across neuron classes. Our case study demonstrates the utility NeuroSC to facilitate exploration of neuronal relationships, leading to new insights on structural features of the connectome and hypotheses for empirical testing.

### Comparisons of NeuroSC to other connectomics atlases

NeuroSC is one of several efforts centered around interpreting the *C. elegans* EM datasets. Other open-source tools for data exploration in *C. elegans* include efforts to capture neuron morphologies and synaptic information (including integration of new connectomes across larval development), to map neurotransmitter and receptor expression, and to record whole brain functional connectivity across genotypes (*Witvliet et al., 2021*; *Altun et al., 2002*; *Cook et al., 2019*; *Fenyves et al., 2020*; *Randi et al., 2023*; *Maitin-Shepard et al., 2021*; *Boergens et al., 2017*). NeuroSC was designed to interface and enhance these existing resources. NeuroSC focuses on comparative, biologically grounded analysis of segmented neuronal features across datasets. This includes new tools for rendering contact profiles, synchronizing 3D views, and navigating C-PHATE diagrams, all of which support multi-scale comparisons of neuronal organization. NeuroSC was designed as a series of modular, open-source tools that can be integrated onto other programs or existing resources, enhancing the existing landscape of connectomic tools. NeuroSC was inspired by tools like NemaNode and WormWiring (*Witvliet et al., 2021*; *Cook et al., 2019*), which enable 3-D visualizations of neuronal morphologies and synaptic sites with synaptic subsets restricted to pre or postsynaptic sites. In NeuroSC, we sought to generate and integrate information beyond the synaptic connectome to include local neuronal regions (contactome) and neuronal morphologies across available developmental vEM datasets.

Contactomes represent features that have been largely overlooked in connectomic datasets, and which capture circuit structures not evident by inspecting solely synaptic relationships (*Brittin et al., 2018*; *Yim et al., 2024*; *Cook et al., 2023*). NeuroSC extends existing representations to also offer user-driven experience with choice over the visualization of specific synaptic sites, the option to search for synaptic partners, and the ability to customize the color of each synaptic representation (*Figure 7*). NeuroSC representations complement resource databases like WormAtlas, which hosts digitized electron micrographs and schematics of neuron morphologies with aggregated information on each neuron (*Altun et al., 2002*). Neuroglancer (*Maitin-Shepard et al., 2021*), developed by Google, is a powerful interactive tool for visualizing large-scale 3D electron microscopy (EM) and other neuroimaging datasets beyond the *C. elegans* community. Similar in functionality to Webknossos (*Boergens et al., 2017*), it provides detailed views of high-resolution EM data and enables users to explore individual datasets effectively, with a strong emphasis on synaptic connectivity. NeuroSC builds upon these capabilities by offering a complementary approach focused on comparative connectomics and broader neuronal relationships. NeuroSC enables the visualization of EM datasets alongside neuron reconstructions while allowing users to compare neuron relationships across multiple carefully curated datasets. These datasets have been standardized across time points and methodologies, making it possible to track developmental changes in neural circuits with confidence. In addition to synaptic connections, NeuroSC highlights the contactome with rendered contact patches, providing a more complete picture of spatial relationships. Users can dynamically adjust colors of rendered objects, generate scale bars for precise comparisons, and clearly visualize synapses with large geometric representations of pre- and post-synaptic structures. Importantly, neuron names and object relationships appear directly when hovering over a rendering, eliminating the need for external lookup tables. NeuroSC also supports downloading models and creating publication-ready figures, making it a valuable tool for both research and presentation. While NeuroSC does not prioritize the display of raw EM data, this information is accessible through https://NeuroSC.cytoshow.org/ (link also available on https://neurosc.net/ in the "About" section) for those who want direct EM inspection. By emphasizing comparability and standardized dataset integration, NeuroSC enables researchers to uncover insights into neural development and connectivity that might not be as easily accessible with single-dataset visualization tools. Together, existing tools and NeuroSC provide complementary ways to explore and analyze complex neural datasets, each offering unique strengths to the neuroscience community.

## NeuroSC design and future directions

NeuroSC code and development was intentional in its design as an open-source resource that is modular and allows integration of additional features and data structures (*Cantarelli et al., 2018*). It is a hypothesis-generating tool that can be equally used by educators seeking to teach neuroanatomical principles and researchers seeking to identify changes across connectome datasets. NeuroSC could be integrated into emerging datasets, including developmental time courses of cell-specific transcriptomic data that would enable further insights on the molecular events underpinning neuronal development—from synaptogenic processes to the logic of neurotransmitter use (*Packer et al., 2019*; *Taylor et al., 2021*; *Fenyves et al., 2020*) and how it sustains functional connectivity (*Randi et al., 2023*). Future iterations of NeuroSC could also include positions and relationships of neurons to non-neuronal cell types, as well as the relative networks of segmented and quantified organelles within cells. NeuroSC could be used to compare new datasets from genetic variants, from animals trained under specific conditions, or from additional developmental datasets across embryogenesis. As such, the pipeline and design of NeuroSC can serve as a sandbox to examine the value of the integration of datasets in exploring representations of neuronal relationships across connectomes. Currently, the addition of new datasets must be performed in-house. In the future, standardized practices for EM generation and annotation data could facilitate the implementation of a standard importer.

NeuroSC forms part of a longer tradition that has leveraged the pioneering datasets generated for *C. elegans* connectomes towards exploring structure-function relationships in the nervous system. While the smaller scale of the *C. elegans* neuropil allowed us to rigorously vet the utility of these approaches, we suggest that these same methods would be beneficial in comparative studies in neuropils of other species, including those with less stereotypically formed connectomes. We recognize that more complex organisms possess orders of magnitude more neurons, with significantly larger neuron populations per cell class. However, our previous work has demonstrated that DC/

CPHATE clustering of *C. elegans* neurons consistently pulls out clusters of shared neuron classes and shared functional roles (*Moyle et al., 2021*). Building on this foundation, we envision applying similar clustering approaches to larger connectomes, aiming to identify classes and functionally related neuronal groups in more complex nervous systems. We suggest that contact profiles, along with neuron morphologies and synaptic partners, can act as 'fingerprints' for individual neurons and neuron classes. These 'fingerprints' can be aligned across animals of the same species to create identities for neurons. Frameworks for systematic connectomics analysis in tractable model systems such as *C. elegans* are critical in laying a foundation for future analyses in other organisms with up to a billion-fold increase in neurons (*Toga et al., 2012*). We think of these collective efforts as akin to the foundational work from *C. elegans* in pioneering genomic analysis and annotations ahead of the Human Genome Project (*Stein et al., 2001*; *Collins and Fink, 1995*). We believe that further integration of datasets in platforms like NeuroSC would be key in determining the representations and features necessary for the interpretation and analyses of other connectomes.

## Materials and methods

### Lead Contact
Further information and requests can be directed to Daniel.colon-ramos@yale.edu.

### Data code and availability
Figures in this article have been generated with NeuroSC (*Figure 5D*; *Figures 6 and 7*, *Figure 5—figure supplement 1G–I*, *Figure 8—figure supplement 2*, *Figure 7—figure supplement 1*, *Figure 8—figure supplement 1A–B*, *Figure 8*, *Figure 8—figure supplements 2–8*; *Figure 4—video 1*, *Figure 6—videos 1; 2*, *Figure 7—video 1*) and CytoSHOW (*Figures 1–4, 5A and C*, *Figure 4—figure supplement 1*, *Figure 8—figure supplement 1C*). Data can be visualized via the viewer at https://neurosc.net/ or by downloading glTF files from NeuroSC and using a glTF viewer to visualize them. Additionally, the data generated for NeuroSC is available in .OBJ file format at (and can be visualized from a local hard drive with CytoSHOW https://NeuroSC.cytoshow.org/). All Excel files for DC iterations and adjacency quantifications can be found in *Supplementary files 3–13*. Tutorials for NeuroSC are available on https://neurosc.net/ upon opening the website, within the main menu of the website (*Figure 8—figure supplement 4*), and in the supplementary materials (*Figure 8—figure supplements 1–8*; *Figure 4—video 1*, *Figure 6—video 2*, *Figure 7—video 1*). These tutorials generally cover the process of engaging in analysis at and across specific developmental stages by filtering the data items and adding items to viewers (*Figure 8—figure supplement 6*). General understanding for how to use C-PHATE to analyze neuronal relationships can be found in *Figures 1 and 4*, *Figure 8—figure supplement 2*; *Figure 4—video 1*, and in our previous publication (*Moyle et al., 2021*). For additional information on filters and in-viewer changes to the data (colors, developmental stages, downloading data), see *Figure 8—figure supplement 1*, *Figure 8—figure supplement 3*, *Figure 8—figure supplement 7*, *Figure 8—figure supplement 8*, and *Figure 6—video 2*, *Figure 7—video 1*. All code for website development is available at GitHub (https://github.com/colonramo-slab/NeuroSCAN, *Emerson, 2025*). and for information on website architecture and data model see *Figure 8—figure supplements 9–10*.

### Experimental model and subject details
Volume electron microscopy (vEM) data and segmentation of neurons and synapses were analyzed from *Witvliet et al., 2021*; *White et al., 1986*; *Brittin et al., 2018*; *Cook et al., 2019*. We analyzed available EM datasets that were transversely sectioned and segmented (*Witvliet et al., 2021*; *Brittin et al., 2021*; *White et al., 1986*). We deleted the CAN neurons in the L1-L3 datasets to keep these datasets consistent with the legacy datasets L4 and Adult (N2U), which do not contain CAN neurons (as in *Moyle et al., 2021*).

### Method details
All 3-D object isosurfaces (Morphologies (Neurons), Contacts, Synapses, C-PHATE plots) were generated from segmented EM datasets using a modified version of the ImageJ 3D viewer plug-in (*Schmid et al., 2010*) implemented in CytoSHOW (cytoshow.org). This tool employs the marching cubes

algorithm for polygon generation. All 3-D objects are first exported as wavefront (.OBJ) files, then converted to GL Transmission Format (.glTF) file format, which does not distort the resolution but compacts the file information to enable faster loading times in the web-based 3-D viewer. We intentionally avoided surface smoothing to renderings to preserve the details of the raw EM data.

## Pixel threshold distance for adjacency profiles and contacts

We identified two challenges in compiling Electron Microscopy (EM) datasets for comparisons: (1) how to uniformly capture neuronal relationships based on areas of physical adjacency (contact) across datasets that have differences in volume depth and in x-y-z resolutions and (2) how to standardize across datasets in which membrane boundaries had been called using a variety of methods, including contrast methods and segmentation methods (hand-drawn vs predicted via centroid node expansion by a shallow convolutional neural network) (*Witvliet et al., 2021*; *Brittin et al., 2018*; *White et al., 1986*). To address this, we first standardized the region of the neuropil across all developmental stages as in *Moyle et al., 2021*. Briefly, all cell bodies were deleted, and we used the entry of the nerve ring neurons into the ventral cord as the posterior boundary landmark for the entire volume, focusing on the AIY Zone 2 (*Colón-Ramos et al., 2007*; slice range *Supplementary file 1*). Previously reported adjacency profiles used 10 pixels (or 45 nm) as the pixel threshold distance for the L4 (JSH) and Adult (N2U) datasets (*Moyle et al., 2021*). To account for differences in resolution (x-y axis) and in calling membrane boundaries between the L4 and Adult datasets and L1-L3 datasets, we designed a protocol to define the pixel threshold for each dataset. In short, for two cells that are in direct contact *Figure 5—figure supplement 1D* in the manually segmented datasets (L4 and Adult), we calculated the length of overlap needed to reach from the segmented edge of one cell, across the membrane, and into the adjacent cell, when the segmented area of one cell is expanded by 45 nm (10 pixels). This results in an average overlap of 30 nm for directly contacting cells in the L4 dataset. Then, in each computationally segmented dataset (L1-L3), we empirically tested the distance (e.g. 55 nm, 60 nm, 62 nm) required to achieve a similar overlap of 30 nm in direct contact cells. That empirical number (in nm) was used for adjacency calculations and rendering of contacts. The numbers were converted from nanometers into pixels to create a pixel threshold distance for each dataset, and these are shown in *Supplementary file 1*. Once these corrections had been applied, we calculated the cell-to-cell adjacency scores for all cell pairs in each dataset by using the measure_adjacency algorithm from https://github.com/cabrittin/volumetric_analysis (*Brittin et al., 2018*; *Supplementary files 8-13*). Adjacency matrices were used for DC (*Brugnone et al., 2019*). The contact surface areas, shown in the contact stats boxes, each represent the sum of the lengths of each all contact outlines for a given cell pair throughout the EM volume, multiplied by the reported z-axis spacing between the slices, giving units of $nm^2$.

## Diffusion condensation

DC is a dynamic, time-inhomogeneous process designed to create a sequence of multiscale data representations by condensing information over time (*Brugnone et al., 2019*). The primary objective of this technique is to capture and encode meaningful abstractions from high-dimensional data, facilitating tasks such as manifold learning, denoising, clustering, and visualization. The underlying principle of DC is to iteratively apply diffusion operators that adapt to the evolving data representation, effectively summarizing the data at multiple scales. The DC process begins with the initialization of an initial data representation, typically the raw high-dimensional data or a preprocessed version. This initial representation is used to construct a diffusion operator, a matrix derived from a similarity matrix that reflects the local geometry of the data. The similarity metric, such as Euclidean distance or cosine similarity, plays a crucial role in defining these local relationships. For contactome datasets, distances between neurons are determined by the pixel overlap between their segmented shapes in the EM dataset. We use these distances to build a graph with weighted edges, in which the weight of the edge represents the pixel overlap (the adjacency in the actual EM segmentation). Affinities between neurons, which are a proxy for their distance in the graph, are then computed as now revised in *Box 1*, Algorithm 1. This process is done iteratively as neurons cluster. Once the initial diffusion operator is established, the algorithm proceeds to the diffusion step. In this step, the diffusion operator is applied to the data, smoothing it by spreading information along the edges of the similarity graph. This operation captures the intrinsic geometry of the data while reducing noise. The specific form of the

## Box 1. Mathematical description of diffusion condensation.

Diffusion condensation
Initialization:
Let $\mathbf{X} = \{x_1, x_2, \ldots, x_n\}$ be the set of $n$ data points in a high-dimensional space. Construct the affinity matrix $\mathbf{A}$, where $A_{ij}$ measures the similarity between $x_i$ and $x_j$. Typically,

$$A_{ij} = \exp\left(-\frac{d(x_i, x_j)^2}{2\sigma^2}\right)$$

for a chosen scale parameter $\sigma$ and distance metric $d$. In the case of the contactome dataset, the data points represent segmented neurons, and affinities are computed using the number of shared pixels shared by pairs of neurons at their boundary.
Diffusion Operator:
Define the degree matrix $\mathbf{D}$ as a diagonal matrix where $D_{ii} = \sum_j A_{ij}$ . Construct the diffusion operator

$$\mathbf{P} = \mathbf{D}^{-1}\mathbf{A}$$

which normalizes the affinity matrix.
Diffusion Step:
Apply the diffusion operator to the data:

$$\mathbf{Y} = \mathbf{PX}$$

This step smooths the data, capturing the intrinsic geometry.
Condensation Step:
After each diffusion step, merge data points that are within a small distance, $\epsilon$ , from each other to form a condensed representation. Specifically, data points $y_i$ and $y_j$ are merged if

$$\|y_i - y_j\| < \epsilon.$$

The threshold $\epsilon$ is computed as a small fraction of the coordinate-wise (element-wise) maximum pairwise distance among all pairs of points after the first diffusion step:

$$\epsilon = \frac{1}{10,000} \cdot \max_{i,j} \|y_i - y_j\|_\infty$$

where $\|\cdot\|_\infty$ denotes the element-wise (coordinate-wise) maximum norm.
Let $\pi : \{1, 2, \ldots, n\} \to \{1, 2, \ldots, k\}$ be a mapping that assigns each data point index $i$ to a cluster index $\pi(i)$ after condensation. The condensed cluster centers are then given by

$$\mathbf{C} = \{c_1, c_2, \ldots, c_k\}, \quad \text{where} \quad c_j = \frac{1}{|\pi^{-1}(j)|} \sum_{i \in \pi^{-1}(j)} y_i.$$

Modularity:
To evaluate the community structure after condensation, compute the modularity score $M$

$$e_M = \frac{1}{2m} \sum_{i,j} \left(A_{ij} - \frac{d_i d_j}{2m}\right) \delta(\pi(i), \pi(j))$$

where:

- $d_i = \sum_j A_{ij}$ is the degree of node $i$,

- $m = \frac{1}{2} \sum_{i,j} A_{ij}$ is the total edge weight in the graph, and
- $\delta(\pi(i), \pi(j)) = 1$ if $\pi(i) = \pi(j)$ (i.e., $x_i$ and $x_j$ belong to the same cluster), and 0 otherwise.

Iteration:
Repeat the diffusion and condensation steps, adjusting the parameter σ adaptively and keeping track of the modularity score, until all points are merged. Output iteration with highest modularity score.

diffusion operator, such as the heat kernel or graph Laplacian, significantly impacts how information is propagated during this step. Following the diffusion step, the condensation step updates the data representation by aggregating diffused data points if the distance between them falls below a 'merge threshold'. This step creates a more compact and abstract representation of the data. These diffusion and condensation steps are iteratively repeated. At each iteration, the diffusion operator is recomputed based on the updated diffuse data representation, ensuring that the process adapts to the evolving structure of the data. The iterations continue until a stopping criterion is met, such as convergence of the data representation to a single point. The output of the DC process is a sequence of multiscale data representations. Each representation in this sequence captures the data at a different level of abstraction, with earlier representations preserving more detailed information and later representations providing more condensed summaries. This sequence of representations can be utilized for various tasks, including manifold learning, denoising, clustering, and visualization. By iteratively smoothing and condensing the data, DC reveals the underlying structure of high-dimensional datasets. The threshold (epsilon) used to merge data points in each iteration is set as a small fraction of the spatial extent of the data: for each coordinate dimension (x, y, z), we compute the range (maximum minus minimum), take the maximum of these three values, and divide it by 10,000. This process is performed iteratively for each round of clustering until all data points cluster into a single point. During DC, we track the modularity of the resulting clusters at each iteration and select the iteration with the highest modularity to define the clusters that represent the strata (*Moyle et al., 2021*; *Brugnone et al., 2019*). Mathematically, modularity is calculated by comparing the actual number of edges within clusters to the expected number of such edges in a randomized network with the same degree distribution (*Newman, 2006*). A higher modularity value implies that nodes within the same cluster are more densely connected to each other than to nodes in other clusters. A detailed algorithm description is provided in *Box 1* and Algorithm 1.

Algorithm 1. **Diffusion condensation**

1: **Input:** Data matrix $\mathbf{X} = x_1, x_2, \ldots, x_n \in \mathbb{R}^{n \times d}$, number of iterations $T$, scale parameter σ, condensation threshold $\epsilon$

2: **Output:** Condensed data matrix $\mathbf{X}_{\text{condensed}}$

3: **Initialize:** Construct affinity matrix $\mathbf{A}$, degree matrix $\mathbf{D}$, and diffusion operator $\mathbf{P}$

4: $\mathbf{A}_{ij} \leftarrow \exp\left(-\frac{d(x_i,x_j)^2}{2\sigma^2}\right)$ for all $i, j$

5: $\mathbf{D} \leftarrow \text{diag}(\sum_j \mathbf{A}_{ij})$

6: $\mathbf{P} \longleftarrow \mathbf{D}^{-1}\mathbf{A}$

7: **for** iteration = 1 to $T$ **do**

8: **Diffusion Step:**
 $\mathbf{Y} \leftarrow \mathbf{PX}$

9: **Condensation Step:**
 Merge data points $x_i$ and $x_j$ if $d(x_i, x_j) < \epsilon$ to form condensed cluster centers $\mathbf{c} = c_1, c_2, \ldots, c_k$

10: $\mathbf{X} \leftarrow \mathbf{C}$

11: **Update:**

12: $\mathbf{A}_{ij} \leftarrow \exp\left(-\frac{d(x_i,x_j)^2}{2\sigma^2}\right)$ for all $i, j$

13: $D \leftarrow \text{diag}(\sum_j \mathbf{A}_{ij})$

14: $\mathbf{P} \leftarrow \mathbf{D}^{-1}\mathbf{A}$

15: **end for**

16: **Return:** $\mathbf{X}_{\text{condensed}} \leftarrow \mathbf{X}$

## C-PHATE

C-PHATE is an extension of the PHATE technique (*Moon et al., 2019*) which is specifically aimed at handling and visualizing high-dimensional biological data. C-PHATE is specifically designed to handle compositional data, which are datasets where the components represent parts of a whole and are inherently constrained. It learns the intrinsic manifold of the data, effectively capturing non-linear relationships and structures that are not apparent with traditional methods like PCA or t-SNE. The C-PHATE algorithm starts by loading affinity matrices associated with specific clustering obtained from DC. These matrices are normalized to generate kernel matrices that emphasize the strength of connections within each cluster. The algorithm then builds a connectivity matrix by integrating these kernel matrices based on cluster assignments over multiple time points. This is achieved by first initializing the matrix with kernel matrices along its diagonal and then filling in off-diagonal blocks with transition probabilities that reflect how clusters transition from one time point to the next. Next, we apply the PHATE dimensionality reduction technique to the connectivity matrix to generate 3D embeddings of the data. These embeddings are derived from multiple iterations of DC, capturing the geometry of the data at various levels of granularity. The resulting coordinates are saved for subsequent analysis. The final step involves visualizing the PHATE results in a 3D graphics tool, CytoSHOW (Java-based; CytoSHOW.org; https://github.com/mohler/CytoSHOW; *Moyle et al., 2021*). The results are plotted in a 3D environment, with functionality enabling rollover labels to display information about clustered cells. This requires cross-referencing output tables from the original data collection. CytoSHOW is an interactive tool that allows for assigning colors and annotations to individual neurons and clusters of interest. A detailed algorithm description is provided in *Box 2* and Algorithm 2. The Python code for C-PHATE allows for user specification of four numerical parameters within the command line, and we used the same set of values for all C-PHATE plots shown in this report (100, 30, 50, 1). The first two integers define the weighting of connectivity between the current condensation step t and previous steps t-1 (weighting = 100) or t-2 (30), respectively, during construction of the connectivity matrix. Values 100 and 30 consistently resulted in a series of plotted clustering trajectories that form a dome-like convergence of paths, enhancing our visual perception of relative relationships and showcasing the super clusters that constitute anatomical strata in the nerve ring neuropil (*Figure 4—video 1*). The reproducibility of the dome shape depends on assigning two specific PHATE parameters (https://

phate.readthedocs.io/en/stable/api.html) to non-default values when calling PHATE, the 't' value is set to 50; the 'randomstate' value is set to 1.

---

Algorithm 2. **C-PHATE**

---

1: **Input:** Output of the Diffusion Condensation algorithm, number of iterations $T$
2: **Output:** Low-dimensional embedding $\mathbf{Y}$
3: **Initialize:** Load affinity matrices and cluster assignments from diffusion condensation output
4: **for** $t = 1$ to $T$ **do**
5: Load affinity matrix $\mathbf{A}_t$ from file
6: Compute degree matrix $\mathbf{D}_t$ where $D_{t_{ii}} = \sum_j A_{t_{ij}}$
7: Normalize to get kernel matrix $\mathbf{K}_t = \mathbf{D}_t^{-1/2}\mathbf{A}_t\mathbf{D}_t^{-1/2}$
8: **end for**
9: $\mathbf{C}_{\text{PHATE}} \leftarrow$ zero matrix with kernel matrices $\mathbf{K}_t$ along the diagonal
10: **for** $t = 1$ to $T - 1$ **do**
11: Compute transition probabilities matrix $\mathbf{P}_{t,t+1}$ for clusters from time $t$ to $t + 1$
12: $\mathbf{P}_{t,t+1}[i,j] \leftarrow \dfrac{\text{Number of points moving from cluster } i \text{ to cluster } j}{\text{Total number of points in cluster } i \text{ at time } t}$
13: Update off-diagonal blocks of $\mathbf{C}_{\text{PHATE}}$ based on $\mathbf{P}_{t,t+1}$
14: **end for**
15: Compute the PHATE embedding $\mathbf{Y}$ from $\mathbf{C}_{\text{PHATE}}$
16: **Return:** $\mathbf{Y}$

---

## Electron microscopy based 3-D models

To make 3-D models of neuron morphologies from vEM datasets, we created Image-J format regions of interest (ROIs) using published segmentation data (*Witvliet et al., 2021*; *White et al., 1986*; *Brittin et al., 2021*). For a given cell, the stack of all sectioned ROIs was then used to draw binary image masks as input to a customized version of the marching cubes algorithm (*Schmid et al., 2010*) to build and save a 3-D isosurface. All steps of this pipeline were executed within the ImageJ-based Java program, CytoSHOW (*Duncan et al., 2019*). Slightly modified versions of this workflow were also followed for: (1) generating cell-to-cell contact ROIs and (2) for generating 3-D representations of synaptic objects. To align the 3-D models from the variously oriented vEM datasets, all surfaces from a given specimen were rotated and resized to fit a consensus orientation and scale. This was achieved by applying a rotation matrix multiplication and scaling factor to all vertex coordinates in isosurfaces comprising each modeled dataset (*Supplementary file 2*). Each 3-D object (morphology, contact, or synapse) was then exported as a Wavefront file (.OBJ) and then web-optimized by conversion to a Draco-compressed .GLTF file. Each neuron was assigned a type-specific color that is consistent across all datasets to enable facile visual comparison. All the original EM annotations that were used to create the representative 3D models in NeuroSC have been preserved and can be accessed, with instructions via the sister app, CytoSHOW (https://NeuroSC.cytoshow.org, https://github.com/mohler/CytoSHOW; *Duncan et al., 2019*).

## Morphologies

Neuron morphologies were linked across datasets for users to visualize changes over time. To enhance 3-D graphics performance without sacrificing gross morphologies, we employed a defined amount of data reduction when building each cell-morphology object. NeuroSC can therefore display multiple (or even all) neurons of a specimen within a single viewer. The number of vertices for a given object was decreased by reducing 10-fold the pixel resolution of the stacked 2-D masks input into the marching cubes algorithm of CytoSHOW.

## Nerve ring

To make a simplified mesh of the overall nerve ring shape, individual neuron ROIs were fused together into a single nerve-ring-scale-stack of image masks. This was used for input to the marching cubes algorithm. The union of all overlapping enlarged neurite ROIs in a vEM section was data reduced (20-fold reduced pixel resolution). This rendered a performance-friendly outer shell of the nerve ring. Importantly, as noted on the website, the nerve ring rendering should be added after all other manipulations have been made to the scene, as the nerve ring will prevent further interactivity with underlying neurons.

## Box 2. Mathematical description of C-PHATE

Given $n$ data points, $\mathbf{X} = \{x_1, x_2, \ldots, x_n\}$, and the diffusion condensation output, consisting of $\mathbf{C}_t = \{c_1, c_2, \ldots, c_k\}$ denoting the merged data points and $\mathbf{A}_t$ denoting the affinity matrix at iteration $t$.

**Kernel Matrix:** For each iteration, $t$, compute the degree matrix $\mathbf{D}$, where $D_{ii} = \sum_j A_{ij}$. Then, normalize the affinity matrix to construct the kernel matrix $\mathbf{K}_t$:

$$\mathbf{K}_t = \mathbf{D}^{-1/2}\mathbf{A}_t\mathbf{D}^{-1/2}$$

**Initial Connectivity Matrix:** Initialize the connectivity matrix $\mathbf{C}_{\text{PHATE}}$ with zeros. Next, populate it with the kernel matrices, $\mathbf{K}_t$, along its diagonal, reflecting self-connections within each cluster at each time point.

**Update Transition Probabilities:** For each pair of adjacent time points $t$ and $t + 1$, compute a transition probability matrix to determine how points transition between clusters $C_t$ and $C_{t+1}$. Each entry $p_{ij}$ in this matrix represents the probability of moving from cluster $i$ at time $t$ to cluster $j$ at time $t + 1$ is calculated by counting the number of points moving from cluster $i$ to cluster $j$ and normalizing by the total number of points in cluster $i$ at time $t$. This can be expressed as:

$$p_{ij} = \frac{\text{Number of points moving from } i \text{ to } j}{\text{Total number of points in cluster } i \text{ at time } t}$$

Use these transition probabilities to populate the off-diagonal blocks of $\mathbf{C}_{\text{PHATE}}$

**Dimensionality Reduction:** Apply the PHATE algorithm to the final connectivity matrix $\mathbf{C}_{\text{PHATE}}$ to obtain the low-dimensional embedding $\mathbf{Y}$:

$$\mathbf{Y} = \text{PHATE}(\mathbf{C}_{\text{PHATE}})$$

**Visualization:** Visualize low-dimensional embedding $\mathbf{Y}$ in CytoSHOW.

## Contacts

To build 3-D representations of neuron-neuron contacts, we captured the degree of overlap when an adjacent cell outline was expanded by the specimen-specific, empirically-defined pixel threshold distance listed in *Supplementary file 1* (see *Figure 5—figure supplement 1*). This was done for each cell outline. This expansion step employs a custom-written method in CytoSHOW that increases the scale of the adjacent outlined region by the pixel threshold distance (*Supplementary file 1*; *Figure 5—figure supplement 1D and E*), while maintaining its congruent shape. The entire collection of captured 2-D contact overlaps (*Figure 5—figure supplement 1C and F*) for each adjacent neuron pair was then reconstructed as a single 3-D object (*Figure 5—figure supplement 1H*). Contact patches shown in NeuroSC are largely reciprocal (e.g. if there is a AIML contact from PVQL then there will be a PVQL contact from AIML), but rarely, 2-D overlap regions may be too small to be reliably converted to 3-D isosurfaces by the marching cubes algorithm, resulting in absence of an expected reciprocal contact model within the collection. Contacts, like cell morphology models, are named to be automatically linked across time-point datasets and to facilitate user-driven visualization of changes over time.

## Synapses

Synaptic positions were derived from the original datasets and segmentations, which annotate synaptic sites in the EM cross-sections (*White et al., 1986*; *Cook et al., 2019*; *Witvliet et al., 2021*). To represent these coordinates in the 3-D segmented neurons, we used Blocks (presynaptic sites), Spheres (postsynaptic sites), and Stars (electrical synapses). The synaptic 3-D objects were placed at the annotated coordinates (*White et al., 1986*; *Cook et al., 2019*; *Witvliet et al., 2021*). Additionally,

the objects were scaled with the scaling factor (*Supplementary file 2*). Synaptic objects were named by using standard nomenclature across all datasets, as explained in *Figure 8—figure supplement 3*.

We note that the L4 and Adult datasets and the L1-L3 datasets were prepared and annotated by different groups (*White et al., 1986*; *Cook et al., 2019*; *Witvliet et al., 2021*). Integration of these datasets reveals nanoscale disagreements in the alignment of the boundaries and synapses. Our representations reflect the original annotations by the authors. Because of these disagreements in annotations, the synapses are not linked across datasets. However, all the original EM annotations that were used to create the representative 3D models in NeuroSC, including the synaptic annotations, have been preserved and can be accessed by the users via the sister app, CytoSHOW, along with detailed user instructions at https://NeuroSC.cytoshow.org.

### Website architecture

The NeuroSC website architecture and data structure were designed to integrate these key user-driven features via a modular platform and linked datasets. The architecture uses Geppetto, an open-source platform designed for neuroscience applications, modularity, and large datasets (*Cantarelli et al., 2018*). Briefly, the architecture is effectively separated into two applications, a front end React/JavaScript bundle that is delivered to the client, rendering the neuron data and assets, and a Golang application that exposes a JSON API, serving the neuron data and assets based on user interactions (*Figure 8—figure supplement 9*). The backend uses a Postgres database to store underlying data (*Figure 8—figure supplement 10*), a Persistent Storage Volume that houses and serves static assets, and a variable number of Virtual Machines to run the front end and backend application code, scaling as needed to accommodate traffic. The User Interface is a React application that allows users to filter, sort, and search through the Neurons so that they can be added to an interactive canvas (*Figure 8—figure supplement 9*). When users add Neurons to a viewer, a .gltf file is loaded in for a given model (Synapses, Neurons, Contacts) at the selected developmental stage (*Figure 8—figure supplement 9*), which can then be manipulated in the 3D environment or layered with other meshes as needed. NeuroSC can be used on common web browsers (e.g. Google Chrome, Safari) and mobile devices.

The underlying data model makes use of tables representing Synapses, Neurons, Contacts, and Developmental Stages. Relationships between these models are represented by foreign keys (*Figure 8—figure supplement 10*). Source data is defined in a file-tree structure containing various assets (such as .gltf files representing various entities), as well as CSVs which store relationships across entities. The directory structure outlines a vertical hierarchy, starting at the developmental stages, then branching downwards onto neuron and synapse data. A Python script is invoked to traverse the directory tree and parse the files, writing to the database accordingly. This configuration enables: (1) verification of the ingested data and (2) quick search times through the datasets to identify related items. Code is version-controlled in GitHub (https://github.com/colonramoslab/NeuroSCAN, *Emerson, 2025*) and deployed through a CI/CD pipeline when updates are committed to the main branch (*Figure 8—figure supplement 9*).

## Acknowledgements

We are grateful for current and former members of the Colón-Ramos lab for their guidance and suggestions, in particular, Agustín Almoril-Porras and Malcom Díaz García for assisting with data formatting, Patricia Chanabá-López and Andrea Cuentas-Condori for feedback on the NeuroSC website, Mayra Blakey for administrative roles in managing contracts for funding distribution, and Ben Clark and Milind Singh for feedback on the paper. We also thank Stephen Larson, Dario Del Piano and Zoran Sinnema (MetaCell) for initial website software development, method reporting and hosting services. We thank Brandi Mattson for editing early paper drafts. We acknowledge Ryan Christensen and Hari Shroff (Janelia Research Campus) and Patrick La Riviere (University of Chicago) for helpful discussions and guidance for the NeuroSC website. We thank the Research Center for Minority Institutions program, the Marine Biological Laboratories (MBL), and the Instituto de Neurobiología de la Universidad de Puerto Rico for providing meeting and brainstorming platforms. DAC-R acknowledges the Whitman Fellows program at MBL for providing funding and space for discussions valuable to this work. Research in DAC-R and WAM labs was supported by NIH grant R24-OD016474. This work was also funded by the NIH/NINDS grant R35 NS132156-01, DP1 NS111778 and R01 NS076558–2.

# Additional information

## Competing interests

Jamie I Emerson: Jamie I. Emerson is affiliated with Bilte Co. Ventura. The author has no financial interests to declare. Smita Krishnaswamy: Reviewing editor, *eLife*. The other authors declare that no competing interests exist.

## Funding

| Funder | Grant reference number | Author |
|---|---|---|
| National Institutes of Health | R24-OD016474 | Daniel A Colón-Ramos |
| National Institutes of Health | R35 NS132156-01 | Daniel A Colón-Ramos |
| National Institutes of Health | DP1 NS111778 | Daniel A Colón-Ramos |
| National Institutes of Health | R01 NS076558-2 | Daniel A Colón-Ramos |

The funders had no role in study design, data collection and interpretation, or the decision to submit the work for publication.

## Author contributions

Noelle L Koonce, Conceptualization, Data curation, Software, Formal analysis, Validation, Investigation, Visualization, Methodology, Writing – original draft, Project administration; Sarah E Emerson, Conceptualization, Software, Formal analysis, Supervision, Validation, Investigation, Visualization, Methodology, Writing – original draft, Project administration, Writing – review and editing; Dhananjay Bhaskar, Conceptualization, Software, Formal analysis, Visualization, Methodology, Writing – original draft, Writing – review and editing; Manik Kuchroo, Methodology; Mark W Moyle, Conceptualization, Methodology; Pura Arroyo-Morales, Nabor Vázquez-Martínez, Validation, Investigation; Jamie I Emerson, Conceptualization, Software, Visualization, Methodology, Writing – review and editing; Smita Krishnaswamy, Conceptualization, Supervision, Methodology; William A Mohler, Conceptualization, Data curation, Software, Formal analysis, Supervision, Validation, Investigation, Visualization, Methodology, Writing – original draft, Writing – review and editing; Daniel A Colón-Ramos, Conceptualization, Supervision, Funding acquisition, Methodology, Writing – original draft, Project administration, Writing – review and editing

## Author ORCIDs

Noelle L Koonce (ID) http://orcid.org/0000-0002-0597-3499
Sarah E Emerson (ID) http://orcid.org/0000-0001-9587-3784
Dhananjay Bhaskar (ID) https://orcid.org/0000-0001-8068-3101
Manik Kuchroo (ID) http://orcid.org/0000-0001-7512-9739
Mark W Moyle (ID) http://orcid.org/0009-0007-8609-0164
Pura Arroyo-Morales (ID) http://orcid.org/0000-0001-5369-6097
Nabor Vázquez-Martínez (ID) http://orcid.org/0000-0003-3602-2802
Jamie I Emerson (ID) http://orcid.org/0009-0008-9333-7484
William A Mohler (ID) https://orcid.org/0000-0003-3586-547X
Daniel A Colón-Ramos (ID) https://orcid.org/0000-0003-0223-7717

Reviewer #1 (Public review): https://doi.org/10.7554/eLife.103977.3.sa1
Reviewer #2 (Public review): https://doi.org/10.7554/eLife.103977.3.sa2
Reviewer #3 (Public review): https://doi.org/10.7554/eLife.103977.3.sa3
Author response https://doi.org/10.7554/eLife.103977.3.sa4

## Additional files

### Supplementary files

Supplementary file 1. Nerve ring regions, resolutions, and pixel threshold distances used to calculate adjacency matrices and to create contact sites for each dataset.

Supplementary file 2. Scaling factors and rotation corrections for 3-D representations of Neurons, Contacts, and Synapses for each dataset.

Supplementary file 3. Stratum 1 (Red) Sankey diagrams of clustered neurons for each Diffusion Condensation iteration in each dataset.

Supplementary file 4. Stratum 2 (Purple) Sankey diagrams of clustered neurons for each Diffusion Condensation iteration in each dataset.

Supplementary file 5. Stratum 3 (Blue) Sankey diagrams of clustered neurons for each Diffusion Condensation iteration in each dataset.

Supplementary file 6. Stratum 4 (Green) Sankey diagrams of clustered neurons for each Diffusion Condensation iteration in each dataset.

Supplementary file 7. Sankey diagrams of AIM, PVQ, and AVF containing clusters for each Diffusion Condensation iteration in each dataset.

Supplementary file 8. L1 (0 hours post hatching) adjacency counts and searchable counter for summed adjacencies. Type the name of a 'Neuron of Interest' (NOI) in the indicated cell to filter for the summed adjacency counts for each contact partner. For each partner, there are two columns: Total number of contacts (number of EM sections NOI and partner are in contact) and Total Weights (summed number of pixels NOI and partner contacts).

Supplementary file 9. L1 (5 hours post hatching) adjacency counts and searchable counter for summed adjacencies. Type the name of a 'Neuron of Interest' (NOI) in the indicated cell to filter for the summed adjacency counts for each contact partner. For each partner, there are two columns: Total number of contacts (number of EM sections NOI and partner are in contact) and Total Weights (summed number of pixels NOI and partner contacts).

Supplementary file 10. L2 (23 hours post hatching) adjacency counts and searchable counter for summed adjacencies. Type the name of a 'Neuron of Interest' (NOI) in the indicated cell to filter for the summed adjacency counts for each contact partner. For each partner, there are two columns: Total number of contacts (number of EM sections NOI and partner are in contact) and Total Weights (summed number of pixels NOI and partner contacts).

Supplementary file 11. L3 (27 hours post hatching) adjacency counts and searchable counter for summed adjacencies. Type the name of a 'Neuron of Interest' (NOI) in the indicated cell to filter for the summed adjacency counts for each contact partner. For each partner, there are two columns: Total number of contacts (number of EM sections NOI and partner are in contact) and Total Weights (summed number of pixels NOI and partner contacts).

Supplementary file 12. L4 (36 hours post hatching) adjacency counts and searchable counter for summed adjacencies. Type the name of a 'Neuron of Interest' (NOI) in the indicated cell to filter for the summed adjacency counts for each contact partner. For each partner, there are two columns: Total number of contacts (number of EM sections NOI and partner are in contact) and Total Weights (summed number of pixels NOI and partner contacts).

Supplementary file 13. Adult (48 hours post hatching) adjacency counts and searchable counter for summed adjacencies. Type the name of a 'Neuron of Interest' (NOI) in the indicated cell to filter for the summed adjacency counts for each contact partner. For each partner, there are two columns: Total number of contacts (number of EM sections NOI and partner are in contact) and Total Weights (summed number of pixels NOI and partner contacts).

MDAR checklist

### Data availability

All datasets analyzed in this study are publicly available. Electron microscopy reconstructions of *C. elegans* developmental connectomes are accessible through the NeuroSC platform (https://neurosc.net/) and CytoSHOW (https://neurosc.cytoshow.org/), with raw data and downloadable files provided in .OBJ and .glTF formats. Supplementary tables containing adjacency matrices, diffusion condensation iterations, and contact statistics are included with this article. All code for NeuroSC development is available at https://github.com/colonramoslab/NeuroSCAN (*Emerson, 2025*). The original

EM datasets were generated and published by *White et al., 1986*, *Brittin et al., 2021* and *Witvliet et al., 2021*, and are publicly available through their respective repositories.

The following previously published datasets were used:

| Author(s) | Year | Dataset title | Dataset URL | Database and Identifier |
|---|---|---|---|---|
| Brittin C, Cook S, Hall DH, Emmons S, Cohen N | 2021 | A multiscale brain map derived from whole-brain volumetric reconstructions | https://doi.org/10.5281/zenodo.4763083 | Zenodo, 10.5281/zenodo.4383277 |
| Ben Mulcahy DW, James MK, Yaron M, Daniel BR, Yuelong W, Yufang L, Xian KW, Rajeev P, Douglas H, Richard SL, Nir S, Andrew CD, Jeff LW, Aravinthan SDT, Mei Z | 2020 | Eight high-resolution electron microscopy volumes of *C. elegans* brains at different stages of development, spanning from birth to adulthood | https://doi.org/10.60533/BOSS-2020-FQI7 | Bossdb.org, 10.60533/BOSS-2020-FQI7 |

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
