## [Editor Report · eLife Assessment]

NeuroSC is an accessible and interactive tool for streamlined observation of neuronal morphology, membrane contact, and synaptic connectivity across developmental stages in the nematode *C. elegans*. This **important** tool relies on **solid** electron microscopy datasets. This resource will be of high interest to *C. elegans* researchers interested in nervous system wiring and circuit function.

---

## [Referee Report · Reviewer #1 (Public review)]

The authors have done a terrific job and addressed the questions raised in my previous review. There are only some minor requests that I have and list below.

---

## [Referee Report · Reviewer #2 (Public review)]

Summary

The past several years has seen publication of both new (Witvliet et al., 2021) and newly analyzed (Cook et al., 2019; Moyle et al., 2021; Brittin et al., 2021) data for the *C. elegans* connectome. The increase in data availability for a single species allows researchers to examine variability due to both stochastic events and due to changes over development. The quantity of these data are huge. To help the community make these data more accessible, the authors present a new online tool that allows examination of 3D models for *C. elegans* neurons in the central neuropil across development. In addition to visualizing the overall structure of the neuronal processes and locations of synapses, the NeuroSC tool also allows users to probe into the C-PHATE visualization results, which this group previously pioneered to describe similarities in neuron adjacency (Moyle et al., 2021).

Strengths

The ability to visualize the data from both a connectomics and contactomics perspective across developmental time has significant power. The original *C. elegans* connectome (White et al., 1986) presented their circuits as line drawings with chemical and electrical synapses indicated through arrows and bars. While these line drawings are incredibly useful, they were necessary simplifications for a 2D publication and lack details of the complex architecture seen within each EM image. Koonce et al takes advantage of their own and others segmented image data of each neuronal process within the nerve ring to create a web interface where users can visualize 3D models for their neuron of choice. The C-PHATE visualization is intended to allow users to explore similarities among different neurons in terms of adjacency and then go directly to the 3D model for these neurons. The 3-D models it generates are beautiful and will likely be showing up in many future presentations and publications. The tool doesn't require any additional downloading and is open source. This revision includes an option where hovering over an individual neurons, synapse, or contact will pull up a statistics panel. The addition of text to the video tutorials in the revision is very useful.

Weaknesses

There are several bugs with this tool, which make it a bit clunky to use and suggest a lack of rigorous testing. There are also issues with data availability. I was disappointed that my "recommendations for the authors", which focused on the user interface, were not addressed in the response to reviewers.

---

## [Referee Report · Reviewer #3 (Public review)]

Summary:

This work provides graphical tools for reconstructing the detailed anatomy of a nervous system from a series of sections imaged by electron microscopy. Contact between neuronal processes can direct outgrowth and is necessary for connectivity, thus function. A bioinformatic approach is used to group neurons according to shared features (e.g., contact, synapses) in a hierarchy of "relatedness" that can be interrogated at each step. In this work, Koonze et al analyze vEM data sets for the *C. elegans* nerve ring (NR), a dense fascicle of processes from181 neurons. In a bioinformatic approach, the clustering algorithm Diffusion Condensation (DC) groups neurons according to similar cell biological features in iterations that remove chunks of differences in feature data with each step ultimately merging all NR neurons in one cluster. DC results are displayed with C-Phate a 3D visualization tool to produce a trajectory that can be interrogated for cell identities and other features at each iterative step. In previous work by these authors, this approach was utilized to identify subgroups of neuronal processes or "strata" in the NR that can be grouped by physical contact and connectivity. Here they expand their analysis to include a series of available vEM data sets across *C. elegans* larval development. This approach suggests that strata initially established during embryonic development are largely preserved in the adult. Importantly, exceptions involving stage specific-specific reorganization of neuronal placement in specific strata were also detected. A case study featured in the paper demonstrates the utility of this approach for visualizing the integration of newly generated neurons into the existing NR anatomy. Visualization tools used in this work are publicly available at NeuroSCAN.

Strengths:

A web-based app, NeuroSCAN, that individual researchers can use to interrogate the structure and organization of the *C. elegans* nerve ring across development.

Weaknesses:

minor revisions

Comments on Revisions:

The authors have satisfactorily addressed my critiques.

---

## [Author Response]

The following is the authors’ response to the original reviews.

**Reviewer #1 (Public review)**
CommentKoonce et al. have generated a web-based visualization tool for exploring *C. elegans* neuronal morphology, contact area between neurons, and synaptic connectivity data. Here, the authors integrate volumetric segmentation of neurons and visualization of contact area patterns of individual neurons generated from Diffusion Condensation and C-PHATE embedding based on previous work from adult volumetric electron microscopy (vEM) data, extended to available vEM data for earlier developmental stages, which effectively summarizes modularity within the collated *C. elegans* contactomes to date. Overall, NeuroSC's relative ease of use for generating visualizations, its ability to quickly toggle between developmental stages, and its integration of a concise visualization of individual neurons' contact patterns strengthen its utility.

We thank that reviewer for this positive assessment of our work.

CommentNeuroSC provides an accessible and convenient platform. However, many of the characteristics of NeuroSC overlap with that of an existing tool for visualizing connectomics data, Neuroglancer, which is a widely-used and shared platform with data from other organisms. The authors do not make clear their motivation for generating this new tool rather than building on a system that has already collated previous connectomics data. Although the field will benefit from any tool that collates connectomics data and makes it more accessible and user-friendly, such a tool is only useful if it is kept up-to-date, and if data formatting for submitting electron microscopy data to be added to the tool is made clear. It is unclear from this manuscript whether NeuroSC will be updated with recently published and future *C. elegans* connectomes, or how additional datasets can be submitted to be added in the future.

We have added new language to more explicitly state the motivations for developing NeuroSC (Introduction, lines 98-111, and discussion lines 375-384). In a new discussion section, we also include comparisons of the features of NeuroSC with other existing tools, like Neuroglancer and Webknossos, (lines 393-417).

Briefly, the functional features of NeuroSC are substantially different (and do not exist) in other web-based tools for navigating EM datasets, including NeuroGlancer. This is because the intended use of NeuroSC is substantially different (and purposefully synergistic) to the intended use, and tools available, in NeuroGlancer.

NeuroGlancer is a versatile tool designed primarily for web-based visualizations and sharing of large EM datasets. NeuroSC was not designed to enable this type of access to the primary EM data (purposefully done because these features were already available through tools like NeuroGlancer).

Instead, the explicit goal of NeuroSC is to provide a platform specifically optimized for examining neuronal relationships across connectomic datasets. NeuroSC builds on the segmentations emerging from programs like NeuroGlancer, but the tools are tailored to explore relationships such as contact profiles in the context of neuronal morphologies and synaptic positions, and across datasets that represent different animals or different developmental stages.

To achieve this, all datasets in NeuroSC were optimized to facilitate comparisons across different connectomes of segmented neuronal features, including: (1) alignment of the neurons that are compared upon the display of the segmentations; (2) synchronization of the 3D windows; (3) implementation of a ‘universal color code’ across datasets for each neuron and relationship for easy visual comparisons; (4) use of the specific neuronal names to label instances of the same cells across all available datasets. The use of precise neuronal names among separate data sets allows integration of these objects with other catalogued datasets, including genomic and neuronal activity profiles.

The formatting and display of the datasets used in NeuroSC was accompanied by the development of new tools including: (1) Rendering of the contact profiles of all neurons in the context of the morphology of the cell and the synapses and (2) C-PHATE diagrams to inspect multidimensional relationship hierarchies based on these contact profiles. In NeuroSC, C-PHATEs can be navigated and compared across multiple stages of development while visualizing neuronal reconstructions, allowing users to compare neuronal relationships across individual datasets.

We agree with the reviewer that these tools are most useful when integrated. With that intention in mind, we designed NeuroSC as a series of modular, open-source tools that could be integrated into other programs, including Neuroglancer. In that sense our intent was not to produce another free-standing tool, but a set of tools that, if useful, could be integrated to other existing web-based connectomic resources to enhance the user experience of navigating complex EM datasets and draw biological meaning from the relationships between the neurons. Additionally, we intentionally designed NeuroSC to enable the ability to integrate new methods of understanding neuron relationships as they arise. We have dedicated a more detailed section to the discussion (lines 369- 417) to better convey this intention and directly address the unique abilities of NeuroSC as a complementary tool to the powerful existing tools, including Neuroglancer.

CommentThe interface for visualizing contacts and synapses would be improved with better user access to the quantitative underlying data. When contact areas or synapses are added to the viewer, adding statistics on the magnitude of the contact area, the number of synapses, and the rank of these values among the neuron's top connections, would make the viewer more useful for hypothesis generation. Furthermore, synapses are currently listed individually, with names that are not very legible to the web user. Grouping them by pre- and postsynaptic neurons and linking these groups across developmental stages would also be an improvement.[what do they even mean by linking?]

We thank the reviewer for this insightful comment and have implemented several improvements to address these suggestions. Specifically, we have added new features to enhance user access to quantitative data within the NeuroDevSCAN viewer:

Cell, Patch, and Synapse Statistics: Users can now see a statistics panel when clicking on a rendered neuron, contact patch, or a synapse. These panels provide the following information, respectively, and are highlighted in lines 303-315:

Cell Stats: Click on a cell rendering to show cell stats which displays the total volume and surface area of the selected neuron within the defined neuropil area of our datasets (see Methods).

Contact Stats: Click on a patch rendering to show ‘contact stats’. This pop up displays quantifications of the selected contact relationship. Rank compares the summed surface area of contacts ("patches") between these two neurons relative to all other contact relationships for the primary neuron for the cell and the whole nerve ring. A rank of 1, for example, means this neuron pair shares the largest contact surface area of the examined relationship. “Total surface area” is displayed in nanometers, and is the summed surface area of all patches of this identity. Contact percentages are presented in two ways: (1) as the proportion of the primary cell's total surface area occupied by the contact in question, and (2) as the proportion of the total surface area of the nerve ring occupied by that same contact. (Showcased in figure S5).

Synapse Stats: A click on a synapse rendering now shows ‘synapse stats’, which displays the number of synapses of the selected identity within the primary neuron, including any polyadic synapse combinations involving the primary neurons. (Showcased in figure S7).

(1) Grouping and Readability Improvements: While individual synapses are still visualized, their display has been improved for legibility. We have condensed the lengthy naming scheme to improve clarity and codified the synapse type by using superscript letters C, E, U to represent chemical, electrical and undefined synapses, respectively. This is explained and shown in figure S7, we added arrows to indicate the directionality of presumed information flow at each synapse.

(2) Developmental Linkage: We can link objects across datasets via cellular identity, but each synapse in the dataset does not yet have an identity attributed to its spatial coordinates, preventing us from linking specific synapses across development beyond their connectivity (ie, that a given synapses connects cell X to cell Y, for instance), also addressed in R1.11.

Together, these improvements substantially enhance the utility of the viewer for hypothesis generation by making key quantitative data readily accessible.

CommentWhile the DC/C-PHATE visualizations are a useful tool for the user, it is difficult to understand when grouping or splitting of cell contact patterns is biologically significant. DC is a deterministic algorithm applied to a contactome from a single organism, and the authors do not provide quantitative metrics of distances between individual neurons or a number of DC iterations on the C-PHATE plot, nor is the selection process for the threshold for DC described in this manuscript. In the application of DC/C-PHATE to larval stage nerve ring strata organization shown by the authors, qualitative observations of C-PHATE plots colored based on adult data seem to be the only evidence shown for persistent strata during development (Figure 3) or changing architectural motifs across stages (Figure 4). Quantitation of differences in neuron position within the DC hierarchy, or differences in modularity across stages, is needed to support these conclusions. Furthermore, illustrating the quantitative differences in C-PHATE plots used to make these conclusions will provide a more instructive guide for users of NeuroSC in generating future hypotheses.

There are several ways to visualize DC outputs, and one way to quantitatively compare DC clustering events of neurons is via Sankey diagrams. To make the inclusion of these resources more clear, we have highlighted them in lines 175-178 (Supplemental Tables 3-6). ‘DC outputs for each strata across animals can also be inspected using Sankey diagrams (Supplemental Tables 3-6). These spreadsheets detail the neuron members at each iteration of DC, allowing the user to derive quantitative comparisons of clustering events.’

As the reviewer points out, DC is a deterministic algorithm that will iteratively cluster neurons based on the similarity of their contact profiles. To better explain the selection process for the threshold, the number of DC iterations and the quantitative metrics between the neurons, we have added new text in the Diffusion Condensation methods section. Briefly:

Number of DC iterations: During diffusion Condensation (DC) we track the modularity of the resulting clusters at each iteration and select the iteration with the highest modularity to define the clusters that represent the strata (Moyle et al., 2021), (Brugnone et al., 2019). Mathematically, modularity is calculated by comparing the actual number of edges within clusters to the expected number of such edges in a randomized network with the same degree distribution (Newman et al., 2006). A higher modularity value implies that nodes within the same cluster are more densely connected to each other than to nodes in other clusters. We now better explain this in lines 562-567.

Threshold for merging points: The threshold (epsilon) used to merge data points in each iteration is set as a small fraction of the spatial extent of the data: for each coordinate dimension (x, y, z), we compute the range (maximum minus minimum), take the maximum of these three values, and divide it by 10,000. This process is performed iteratively for each round of clustering until all data points cluster into a single point. We have updated the manuscript to clarify this threshold selection and included this information in the revised algorithm description and pseudocode. We now better explain this in lines 556-559.

Distances between neurons in DC C-PHATE: In our previous description in Box 1 algorithm 1, we had provided a general algorithm for DC for any high dimensional dataset. We have now revised the algorithm to indicate how we used DC for these EM datasets.

Distances between neurons are determined by the pixel overlap between their segmented shapes in the EM dataset. We use these distances to build a graph with weighted edges, in which the weight of the edge represents the pixel overlap (the adjacency in the actual EM segmentation). Affinities between neurons, which are a proxy for their distance in the graph, are then computed as now revised in Box 1, Algorithm 1. This process is done iteratively as neurons cluster. To better communicate this, we have changed the text in lines 533-538.

CommentR1.5. While the case studies presented by the authors help to highlight the utility of the different visualizations offered by the NeuroSC platform, the authors need to be more careful with the claims they make from these correlative observations. For example, in Figure 4, the authors use C-PHATE clustering patterns to make conclusions about changes in clustering patterns of individual neurons across development based on single animal datasets. In this and many other cases presented in this study with the limited existing datasets, it is difficult to differentiate between developmental changes and individual variability between the neurite positions, contacts, and synapse differences within these data. This caveat needs to be clearly addressed.

We now better explain in the manuscript that the selected case study, of the AVF neuron outgrowth, is not one of just correlation based solely on an EM dataset. Instead, the case study represents the NeuroSC-driven exploration of a biologically significant event supported by several independent datasets, as now explained in lines 257-276.

Briefly, we agree with the reviewer that examining differences across individual EM datasets is insufficient evidence to make conclusions about developmental changes. But the strength of NeuroSC is in its ability to combine and compare multiple datasets, bolstering observations that are not possible by looking at just one dataset, and providing new insights on the way to new hypotheses. We now better explain that we are not looking at single connectomes in isolation and then deriving conclusions, but instead using NeuroSC to compare across 9 EM datasets. We better explain how the tools in NeuroSC, including C-PHATE, enabled comparisons across these multiple connectomes to identify apparent differences in neuronal relationships. We then explain that by using NeuroSC, we could examine these variations in neuronal relationships at the level of individual, cell biological differences of neuronal morphologies between the developmental datasets. This could be due, as pointed by the reviewer, to differences due to development, or just differences between individual animals. In the case of AVF, that features are absent in all early specimens, then arise and persist in all specimens after a certain time point, which lead us to hypothesize they result from a developmental event. Because the segmented objects in NeuroSC are linked to neuronal identities, we are also able to cross reference our observations from the EM datasets with information in other datasets and the literature. In the specific case of postembryonic development of AVF outgrowth, we can now tie the knowledge, from developmental lineage information and molecular profiles, that AVF is a postembryonically born neuron (Sulston et al. 1977, Sun et al 2022, Poole et al 2024, wormatlas.org) to the outgrowth dynamics of its neurites using the postembryonic EM datasets. Our findings using NeuroSC provide a proof of concept of the utility of the resource and extended our understanding of how the outgrowth of this neuron affects the relationships between the neural circuits in the nerve ring.

CommentR1.6. Given that recent studies have also quantified contact area between neurons across multiple connectomes (Cook et al., Current Biology, 2023; Yim et al., Nature Communications, 2024), and that the authors use a slightly different approach to quantify contact area, a direct comparison between contact area values obtained in this study with prior studies seems appropriate.

We acknowledge that there are multiple different approaches to calculate adjacencies. In the papers cited above, there are 3 different algorithms used:

(1) Brittin 2019 (python parse Track EM, boundary thresholds), used in Cook et al 2023, Moyle 2021, and this study.

(2) Witvliet 2021 (Matlab 2D masks), used in Cook et al 2023.

(3) Yim 2024 (3D masks), used in Yim et al 2024.

To briefly describe the different approaches, and the methods we chose for this paper:

Algorithm 1 (used in this study) defines adjacency based on distances between boundary points in TrakEM2 segmentations, allowing threshold tuning to accommodate differences in resolution and image quality across datasets—an important feature for consistent cross-dataset comparisons.

Algorithm 2 infers contact via morphological dilation of VAST segmentations, identifying adjacency through overlapping expanded boundaries.

Algorithm 3 uses voxelwise contact detection with directional surface area measurements and normalization to account for dataset size differences.

In NeuroSC, we use algorithm 1, mostly because we had tested the rigor of this method in (Moyle et al. 2021), where we have shown that results were robust across a range of thresholds. This flexibility enables tailored application across datasets of varying quality and scale, critical for NeuroSC’s mission of curating data sets across differing methodologies to allow for direct relationship comparisons. We detail the methodology for defining thresholds for each dataset in methods section lines 492-521, defined in Supplementary table 1. Another difference between our analysis and the previously cited work is that for our analysis we also chose to include all individually resolved neurons, including post-embryonic cells, without collapsing them into left/right or dorsal/ventral symmetry classes. In this way our approach retains the full cellular resolution of the nervous system.

CommentNeuroglancer is not mentioned at all in the manuscript, despite it being a very similar and widely accepted platform for vEM data visualization across model organisms. An explicit comparison of NeuroSC and Neuroglancer would be appropriate, given the similarity of the tools. Currently, published *C. elegans* data (Witvliet et al., 2021; Yim et al., 2024) use Neuroglancer-based viewers, and directly comparing NeuroSC and highlighting its strengths relative to Neuroglancer would strengthen the paper.

In the original manuscript we had not mentioned tools like Neuroglancer because we envisioned them as distinct, in intended use and output, from NeuroSC. But, as explained in R1.2 comment, in the revised version we have included a section in the Introduction lines 98-108 and in the Discussion (lines 369- 417) that compares these types of web-based tools and highlights synergies.

CommentAssigning shorthand names to strata, such as "shallow reflex circuit" (page 4, line 172), may oversimplify this group of neurons. Either more detailed support for shorthand names of C-PHATE modules should be included, or less speculative names for strata should be used.

We appreciate this comment and understand that the original language used in the manuscript to describe strata categorizations may run the risk of oversimplification. We have now clarified the text to communicate that: (1) Strata are labeled by numbers (Strata 1, Strata 2, Strata 3 and Strata 4), rather than functional features of the neurons forming part of the strata, and that (2) the assignment of ‘strata’ is just one level of classification available via DC/CPHATE (as explained below).

To be sure, we have observed and published (Moyle et. al. Nature 2021) that within a given stratum, many neurons share the functional identities that we have used as summary descriptors for the strata (eg, shallow reflex circuits for Stratum 1; sensory and integrative circuits in Strata 3 and Strata 4; command interneurons in Strata 2, etc). However, those cell types are not the only members of the strata. We have adjusted the language in lines 197-204 to reflect this more clearly. “Stratum 1, which contains most neurons contributing to shallow reflex circuits that control aversive head movements in response to noxious stimuli, displayed the fewest changes among the developmental connectomes (Figure 3B–F; Supplementary Table 3). In contrast, *C. elegans* exhibit tractable behaviors that adapt to changing environmental conditions (Flavell et al., 2020). Strata 3 and 4 contain most neurons involved in circuits associated with such learned behaviors, including mechano- and thermo-sensation. This is reflected in Strata 3 and 4 showing the most change in neuronal relationships across postembryonic development.“

CommentThe authors state that NeuroSC can be applied to other model organisms. Since model organisms with greater neuron numbers include more individual neurons per cell class, the authors should support this by quantitatively demonstrating how DC/C-PHATE relationships correlate with shared functional roles among *C. elegans* neurons.

We now clarify in the manuscript that, like in other organisms, *C. elegans* neurons are also grouped into functional classes with shared characteristics. In the context of the cylindrical nerve ring of the animal, these neuronal classes are sometimes bilaterally symmetric (forming left-right pairs), four-fold symmetric and six-fold symmetric. We now explain in the discussion that the DC/CPHATE analyses group these neuron classes and their relationships (lines 442-451). In the specific section mentioned by the reviewer, we now also add new text to contextualize this concept and how it might relate to the possible use of these tools in organisms with larger nervous systems: ‘However, our previous work has demonstrated that DC/CPHATE clustering of *C. elegans* neurons consistently pulls out clusters of shared neuron classes and shared functional roles Moyle et al. (2021). Building on this foundation, we envision applying similar clustering approaches to larger connectomes, aiming to identify classes and functionally related neuronal groups in more complex nervous systems. We suggest that contact profiles, along with neuron morphologies and synaptic partners, can act as ‘fingerprints’ for individual neurons and neuron classes. These ‘fingerprints’ can be aligned across animals of the same species to create identities for neurons. Frameworks for systematic connectomics analysis in tractable model systems such as *C. elegans* are critical in laying a foundation for future analyses in other organisms with up to a billion-fold increase in neurons (Toga et al., 2012).’

CommentLack of surface smoothing in NeuroSC leads to processes sometimes appearing to have gaps, which could be remedied by smoothing with a surface mesh.

We thank the reviewer for the suggestion, and understand the visibility of gaps in certain neuron processes can be distracting. But this was an intentional choice, with our main goal being to show the most accurate representation of the available data segmentation and avoid any rendering interpretations. In this way, we render the data with the highest fidelity we can and as close as possible to the ground truth of the EM segmentation. We have added language to describe this in the methods, lines 490-491, and in Figure legend 5b.

CommentToggling between time points while maintaining the same neurons and contact area in NeuroSC is a really valuable feature. The tool would be improved even more by extending this feature to synapses, specifically by allowing the user to add an entire group of synapses to the viewer at once (e.g. "all synapses between AIM and PVQ"), and to keep this synapse group invariant when toggling between developmental stages.

We thank the reviewer for this suggestion. In response we have now implemented a new feature to ‘clone’ a rendered scene across time while preserving the original elements to ease comparisons. Once the user has rendered a scene, they can use the in-viewer developmental slider to clone the renderings and assigned colors, but display the renderings of the newly selected timepoint. These renderings populate a new window tab which can be dragged to align developmental stage windows side by side. We have added a sentence to account for this in lines 315-317 and to the legend of supplemental Figure S11.

**Reviewer #2 (Public review)**
CommentThe ability to visualize the data from both a connectomics and contactomics perspective across developmental time has significant power. The original *C. elegans* connectome (White et al., 1986) presented their circuits as line drawings with chemical and electrical synapses indicated through arrows and bars. While these line drawings remain incredibly useful, they were also necessary simplifications for a 2D publication and they lack details of the complex architecture seen within each EM image. Koonce et al take advantage of segmented image data of each neuronal process within the nerve ring to create a web interface where users can visualize 3D models for their neuron of choice. The C-PHATE visualization allows users to explore similarities among different neurons in terms of adjacency and then go directly to the 3D model for these neurons. The 3D models it generates are beautiful and will likely be showing up in many future presentations and publications. The tool doesn't require any additional downloading and is open source.

We thank that reviewer for this positive assessment of our work.

CommentWhile it's impossible to create one tool that will satisfy all potential users, I found myself wanting to have numbers associated with the data. For example, knowing the number of connections or the total surface area of contacts between individual neurons wasn't possible through the viewer, which limits the utility of taking deep analytical dives. While connectivity data is readily accessible through other interfaces such as Nemanode and WormWiring, a more thorough integration may be helpful to some users.

We thank the reviewer for this feedback and in response have now implemented displays with quantitative information in NeuroSC. Now, upon hovering over a contact patch or synapse, the user will see the quantitative data of the relationship. For contact patches, you will see the total area shared between two neurons in that dataset. On hovering over a synapse, you will see how many synapses there are in total with the same members and throughout the dataset. We agree that this improves user analyses, (see also R1.3 response).

CommentThere were several issues with the user interface that made it a bit clunky to use. For example, as I added additional neurons to the filter search box, the loading time got longer and longer. I ran an experiment uploading all of the amphid neurons, one pair at a time. Each additional neuron pair added an additional 5-10 seconds to the loading. By the time I got to the last pair, it took over a minute to load. Issues like these, some of which may be unavoidable given the size of the data, could be conveyed through better documentation. I did not find the tutorial very helpful and the supplementary movies lacked any voiceover, so it wasn't always clear what they were trying to show.

We appreciate that some of the more complex models can take a while to load. One of our core goals is to keep the high resolution of our models to most accurately represent the EM data, so we had to compromise between resolution and loading times. But to address this concern we have now added a ‘loading’ prompt that reassures the user when there is a wait. We also added, as suggested, text guidance throughout all of the supplemental videos (Supplemental Videos 1-4).

**Reviewer #3 (Public review)**
CommentA web-based app, NeuroSC, that individual researchers can use to interrogate the structure and organization of the *C. elegans* nerve ring across development In the opinion of this reviewer, only minor revisions are required.

We thank that reviewer for this positive assessment of our work.

CommentContact is defined by length, why not contact area? How are these normalized for changes in the overall dimensions of neurons during development?

To clarify our methodology: the adjacency algorithm that we use generates a 2D adjacency profile by summing the number of adjacent boundary points per EM section, which are then summed across all EM z slices.

Contact area can be derived by multiplying the adjacency length in each slice by pixel resolution and z-thickness. Prompted by the reviewer we have now also calculated and display contact surface areas, along with their ranks among all contact relationships for a given neuron. These can be inspected directly via the interface by clicking on a rendered cell or contact patch (Figure S5 and lines 308-312). We believe these additional surface area metrics enhance the interpretability and utility of the viewer.

We apply normalization at the level of the adjacency threshold to account for dataset-specific differences such as contrast, boundary definition, and age-related changes in neuropil packing density. This normalization is applied before running the adjacency algorithm. We do not normalize by individual neuron size, as the contact data are intended to reflect relational differences between neurons, rather than absolute morphological scaling. In fact, our addition of a scale-spheroid within each rendered model emphasizes the large increase in spatial scale that the nerve ring experiences during larval growth.

CommentFigure 1, C&D, explanation unclear for how the adjacency matrix is correlated with C-Phate schematic in D.

We thank the reviewer for the comment and have clarified this section by adding greater detail to the explanation of how an adjacency matrix is computed (lines 149-155), as well as a description now in the figure legend 1C. Additionally, we revised Figure 1C and D to simplify neuron representations/colors and to simplify the adjacency heat map gradient. We also extended the area of contact between neurons on Figure 1C to better reflect what would be considered a “contact”. Lastly, in the figure, we changed the color and placement for the z plane arrow and label from black to white, to make it more visible, to highlight the method of computing adjacency for each z slice.

CommentFigure 4, panels F & G, unclear why AVF is shown in panel G (L3) but not panel F (L1). Explanation (see below) should be provided earlier, i.e., AVF is not generated until the end of the L1.

We have now clarified this important point by adding labels to Figure 4 panels F and G, ‘Pre-AVF outgrowth’ and ‘Post-AVF outgrowth’ respectively. Briefly, the point is that AVF grows into the nerve ring after the L2 stage, and that is why it is absent in panel F (L1 stage, now with the label ‘Pre-AVF outgrowth’).

CommentLine 146 What is the justification for the statement: "By end of Larval Stage 1 (L1), neuronal differentiation has concluded...."? This statement is confusing since this sentence also states that "90% of neurons in the neuropil...have entered the nerve ring..." which would suggest that at least 10% additional NR neurons have NOT fully differentiated.

We have fixed this sentence in the text. Now the sentence reads ‘By Larval stage 1 (L1) 90% of the neurons in the neuropil (161 neurons out of the 181 neurons) have grown into the nerve ring and adopted characteristic morphologies and positions.

Lines 171-175 What is meant by the statement that "degree of these changes mapped onto...plasticity? What are examples of "behavioral plasticity?"

We have added the following new lines of text (lines 200-204) and now additionally cite a review discussing *C. elegans* behaviors to clarify and give context to behavioral plasticity. ‘*C. elegans* exhibit tractable behaviors which can adapt due to changing environmental conditions (Flavell et. al. Genetics 2020). Strata 3 and 4 contain most neurons belonging to circuits associated with such learned behaviors, including chemo, mechano and thermo sensation. This is seemingly reflected by strata 3 and 4 harboring the most readily recognized set of changes in neuronal relationships across postembryonic development.’

CommentLines 189-190 The meaning of this sentence is unclear, "The logic in....merge events."

This sentence has been deleted and we have instead refocused our descriptions of C-PHATES comparisons by neuronal clustering trajectories and cluster members (rather than iterations).

CommentLines 193-208 This section reports varying levels of convergence across larval development in C-Phate maps for the interneurons AIML and PVQL. Iterations leading to convergence varied: 16 (L1), 14 (L2), 22 (L3), 20 (l4), 14 (adult). The authors suggest that these differences are biologically significant and reflect the reorganization of AIML and PVQL contact relationships especially between the L4 and adult. Are these differences in iterations significant?

We agree this could be confusing and instead of focusing on comparing the iteration at which each merging event occurs, we now focus on examining the differences in members of clusters, before and after the merge event. Cluster membership is easier to interpret than the differences in the number of DC iterations (lines 224-229).

Lines 240-241 States that AVF neurons "terminally differentiate in the embryo" which is not correct. AVF neurons are generated from neuronal precursors (P0 and P1) at the end of the L1 stage which accounts for their outgrowth into the NR during the L2 stage.

We thank the reviewer for the correction and have edited the text to read: ‘AVF neurons are generated from neuronal precursors (P0 and P1) at the end of the L1 stage Sulston et al. (1983); Sun and Hobert (2023); Poole et al. (2024); Hall and Altun (2008); Sulston and Horvitz (1977). AVF neurons do not grow into the nerve ring until the L2 stage, and continue to grow until the Adult stage (lines 261-266).’

CommentLines 289-315. A detailed and highly technical description of website architecture would seem more appropriate for the Methods section.

We agree and have moved this section to the methods as suggested (lines 663-690).

CommentLine 307 "source data is" should be "source data are"

Thank you- we have fixed this grammatical error.

CommentLine 324 "circuits identities" should be "circuit identity".

Thank you- we have fixed this grammatical error.

CommentTrademark/copyright conflict with these sites? https://compumedicsneuroscan.com/about/
https://www.neuroscanai.com/

We thank the reviewer for drawing our attention to this. To avoid potential conflicts, we have proactively altered the name to NeuroSC throughout the paper.